# Realizing long-cycling all-solid-state Li-In‖ TiS$_2$ batteries using Li$_{6+x}$M$_x$As$_{1-x}$S$_5$I (M=Si, Sn) sulfide solid electrolytes

Pushun Lu[1,2], Yu Xia[3], Guochen Sun[1,2], Dengxu Wu[1,2], Siyuan Wu[1,2], Wenlin Yan[1,2], Xiang Zhu[4,5], Jiaze Lu[1,2], Quanhai Niu[5], Shaochen Shi [ID][3], Zhengju Sha[3], Liquan Chen[1,2,5,6], Hong Li [ID][1,2,4,5,6] ✉ & Fan Wu [ID][1,2,4,5,6] ✉

Inorganic sulfide solid-state electrolytes, especially Li$_6$PS$_5$X (X = Cl, Br, I), are considered viable materials for developing all-solid-state batteries because of their high ionic conductivity and low cost. However, this class of solid-state electrolytes suffers from structural and chemical instability in humid air environments and a lack of compatibility with layered oxide positive electrode active materials. To circumvent these issues, here, we propose Li$_{6+x}$M$_x$As$_{1-x}$S$_5$I (M=Si, Sn) as sulfide solid electrolytes. When the Li$_{6+x}$Si$_x$As$_{1-x}$S$_5$I (x = 0.8) is tested in combination with a Li-In negative electrode and Ti$_2$S-based positive electrode at 30 °C and 30 MPa, the Li-ion lab-scale Swagelok cells demonstrate long cycle life of almost 62500 cycles at 2.44 mA cm$^{-2}$, decent power performance (up to 24.45 mA cm$^{-2}$) and areal capacity of 9.26 mAh cm$^{-2}$ at 0.53 mA cm$^{-2}$.

All-solid-state batteries (ASSBs) are considered as the next-generation energy-storage technology due to their safety aspects and theoretically higher energy density than commercial lithium-ion batteries[1–4]. Replacing the flammable and non-aqueous organic liquid electrolyte solutions with more thermal-stable solid electrolytes (SEs) could overcome the safety issues and break through the energy density bottleneck by coupling with lithium metal anode[5]. Among all SEs investigated, sulfides have the highest ionic conductivity at 25 °C[6] and desirable mechanical property for intimate contacts with electrode materials by cold-pressing without using additional liquids or polymer electrolytes[7–10].

Since 1981, various types of highly ionic conductive sulfide SEs have been reported, including Li$_7$P$_3$S$_{11}$ (17 mS cm$^{-1}$ at 25 °C) glass-ceramics[11], Li$_{10}$GeP$_2$S$_{12}$ (LGPS, 12 mS cm$^{-1}$ at 25 °C), Li$_{9.54}$Si$_{1.74}$P$_{1.44}$S$_{11.7}$Cl$_{0.3}$ (LSPSC, 25 mS cm$^{-1}$ at 25 °C)[12,13], Li$_{5.5}$PS$_{4.5}$Cl$_{1.5}$ (10.2 mS cm$^{-1}$ at 25 °C), and Li$_{6.6}$Si$_{0.6}$Sb$_{0.4}$S$_5$I (24 mS cm$^{-1}$ at 25 °C) argyrodites[14,15]. However, most of these reported ionic conductivities were measured by using sintered or hot-pressed pellets with high-temperature heat treatment, which are not convenient for practical application of ASSBs and may facilitate undesirable interfacial reactions[16]. Moreover, these sulfide SEs suffer from severe air instability to induce toxic H$_2$S gas release, rapid structure degradation and performance decay, which further limits their large-scale application in sulfide ASSBs[17]. Recent experimental results reveal that the partial substitution of soft acids including Ge$^{4+}$ [18], Sn$^{4+}$ [19], As$^{5+}$ [20], Sb$^{5+}$ [21], In$^{3+}$ [22] for hard acid P$^{5+}$ can enhance the air stability of phosphorus-based sulfide SEs, based on the hard and soft acid base (HSAB) theory. However, these phosphorus-based sulfide SEs still suffer from irreversible structure degradation and H$_2$S gas release. Phosphorus-free Thio-LISCON (lithium ion superionic conductor) sulfide SEs, such as Li$_4$SnS$_4$[23], 0.4LiI-0.6Li$_4$SnS$_4$[24], Li$_{4-x}$Sn$_{1-x}$As$_x$S$_4$[20,25], Li$_{4-x}$Sn$_{1-x}$Sb$_x$S$_4$[26,27], and Li$_3$SbS$_4$[28], have been designed through complete substitution and exhibit moisture stability and even recoverability after heat treatment. It is noteworthy that a softer acid may not definitely lead to a better moisture stability. The moisture stability of sulfide SEs with these soft acids as the only central cations follows the order of In$^{3+}$ > As$^{5+}$ > Sn$^{4+}$ >

[1]Institute of Physics, Chinese Academy of Sciences, Beijing 100190, China. [2]School of Physical Sciences, University of Chinese Academy of Sciences, Beijing 100049, China. [3]Beijing ByteDance Technology Co Ltd, Beijing 100098, China. [4]Nano Science and Technology Institute, University of Science and Technology of China, Suzhou 215123, China. [5]Tianmu Lake Institute of Advanced Energy Storage Technologies, Liyang 213300 Jiangsu, China. [6]Yangtze River Delta Physics Research Center, Liyang 213300 Jiangsu, China. ✉e-mail: hli@iphy.ac.cn; fwu@iphy.ac.cn

$Ge^{4+} > Sb^{5+}$, according to the thermodynamics analysis[29]. However, the ionic conductivities at 25 °C of this kind of SEs are generally lower than 1 mS cm⁻¹. Recently, Zhou et al.[15] developed phosphorus-free argyrodites, $Li_{6+x}M_xSb_{1-x}S_5I$ (M=Si, Ge, Sn), which displayed high conductivity over 10 mS cm⁻¹ at 25 °C for cold-pressed pellet and improved air stability[30] compared with $Li_6PS_5I$. However, no striking electrochemical performance has been achieved for thioantimonate ASSBs which excessively rely on high ionic conductivity, but compromising interfacial stability[31]. Lately, we reported the potential of $Li_{6.8}Si_{0.8}As_{0.2}S_5I$ (LASI-80Si) solid electrolyte that enables conversion-type cathode $FeS_2$ with high areal capacity and wide-temperature (−60 °C–60 °C) operation capability[32]. However, the ion-conduction mechanisms, air stability, and compatibility with intercalation-type cathode for LASI-80Si are unrevealed.

Here, we report the synthesis and characterizations of various $Li_{6+x}M_xAs_{1-x}S_5I$ (M=Si, Sn) sulfide SEs. The ionic conductivity of $Li_{6+x}M_xAs_{1-x}S_5I$ family can be tuned by adjusting the doping concentration, reaching 10.4 mS cm⁻¹ at 25 °C for the known SE $Li_{6.8}Si_{0.8}As_{0.2}S_5I$ (LASI-80Si)[32]. This value is three orders higher than that of parent phase $Li_6AsS_5I$ (LASI), since the aliovalent substitution introduces additional Li⁺ ions to fall into high-energy lattice sites (i.e., Li3 and Li4 sites). This opens up an inter-cage migration channel and activates concerted migration, which substantially lowers the activation energy from 0.69 to 0.17 eV. Moreover, LASI-80Si has better air stability than phosphorus-based lithium argyrodites $Li_6PS_5X$ (X=Cl, Br, I), due to the tight bonding between soft acid $As^{5+}$ and soft base $S^{2-}$, and the competitive reaction triggered by $Li_2S$ and LiI functional phases[32]. Furthermore, the high ionic conductivity, interfacial passivation effect and Li⁺-supplement effect induced by $Li_2S$ and LiI[32] jointly contribute to improve the electrochemical energy storage performances of LASI-80Si in Li-In||TiS₂ all-solid-state cell configuration at 30 °C, including high specific capacity (222.3 mAh g⁻¹ at 0.76 mA cm⁻²), initial

coulombic efficiency (CE, 99.06% at 0.76 mA cm⁻²), long cycle life (almost 62,500 cycles at 2.44 mA cm⁻²), decent power performance (up to 24.45 mA cm⁻²), high electrode mass loadings (e.g., 44.56 mg cm⁻²) and areal capacity (e.g., 9.26 mAh cm⁻² at 0.53 mA cm⁻²). In addition, replacing the majority part of As by Si also significantly reduces the cost and toxicity of $Li_{6+x}M_xAs_{1-x}S_5I$ family, which is critical for practical application and mass-production.

## Results and discussion
### Elemental substitution and superionic conductivity
In this work, instead of the solid-state time-consuming synthesis method proposed by Zhou et al.[15] for Sb-based argyrodites, $Li_{6+x}M_xAs_{1-x}S_5I$ (M=Si, Sn) are synthesized here through ball-milling and thermal annealing, following the same procedure[32,33] reported for the preparation of $Li_6PS_5X$ (X = Cl, Br, I) and LASI-80Si. Moreover, lightweight, low-cost, and non-toxic Si is used here to substitute the majority of As. As shown in Fig. 1a (x = 0, bottom), X-ray diffraction peaks of the as-synthesized $Li_6AsS_5I$ (LASI) correspond well with the simulated patterns, indicating its successful preparation. However, the ionic conductivity of LASI is only 3.92 × 10⁻⁶ S cm⁻¹ at 25 °C, slightly higher than that of $Li_6PS_5I$ (LPSI, 3.33 × 10⁻⁶ S cm⁻¹ at 25 °C) synthesized with the same procedure (Supplementary Table 1). By replacing P with M (M=Si, Ge, Sn), the ionic conductivity of LPSI can be improved from 10⁻⁶ S cm⁻¹ to 10⁻³ S cm⁻¹ at 25 °C[34,35]. This enhancement is ascribed to the local disorder of I⁻/S²⁻ and the increase of Li⁺ concentration induced by the aliovalent cation substitution. Therefore, Sn/Si substitution is expected to improve the ionic conductivity of LASI, while Ge is not preferable given its high price[36]. It is interesting to note that even at low substitution proportions (x = 0.05, 0.1), LiI and $Li_2S$ functional phases still exist as impurity phases (Fig. 1a, b), which has also occurred for $Li_{6+x}M_xSb_{1-x}S_5I$ (M=Si, Ge, Sn)[15,30]. As the substitution proportion of Sn/Si increases, the amount of $Li_2S$ and LiI also

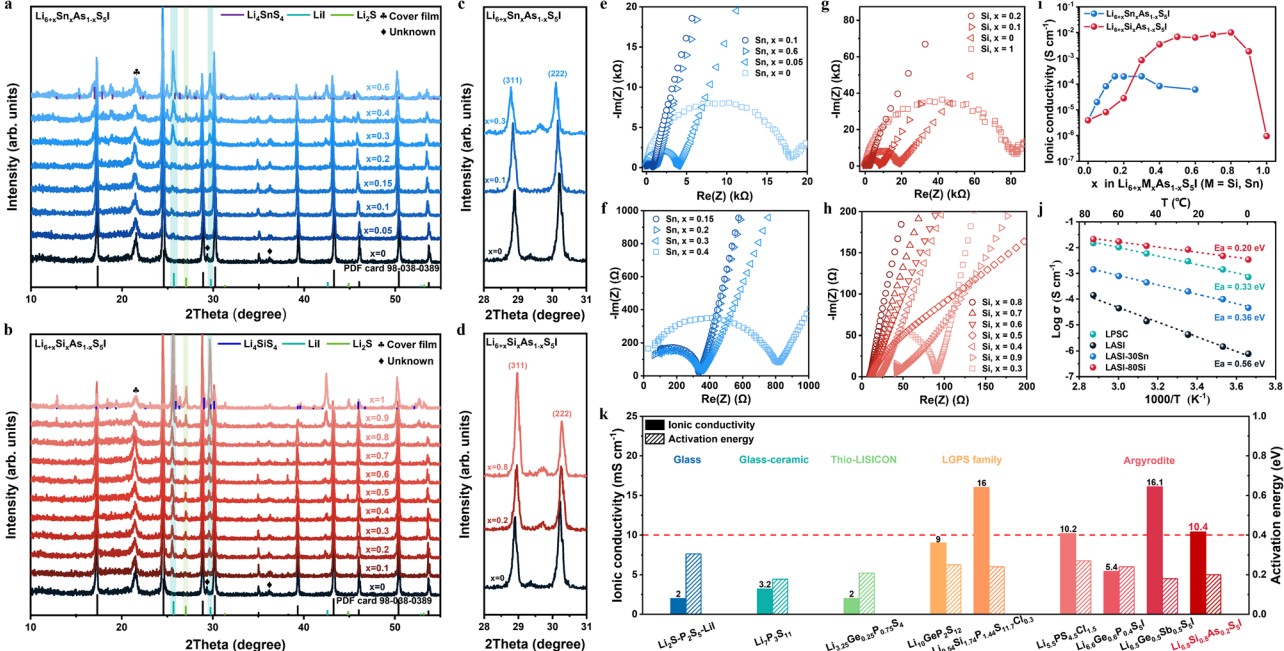

**Fig. 1 | Structural and electrochemical characterization of the synthesized solid-state electrolytes. a–d** X-ray diffraction patterns of **a** $Li_{6+x}Sn_xAs_{1-x}S_5I$ (x = 0, 0.05, 0.10, 0.15, 0.20, 0.30, 0.40, 0.60), **b** $Li_{6+x}Si_xAs_{1-x}S_5I$ (x = 0, 0.10, 0.20, 0.30, 0.40, 0.50, 0.60, 0.70, 0.80, 0.90, 1.00), and the enlarged X-ray diffraction patterns at 28° ~ 31° for **c** $Li_{6+x}Sn_xAs_{1-x}S_5I$ (x = 0, 0.10, 0.30), **d** $Li_{6+x}Si_xAs_{1-x}S_5I$ (x = 0, 0.20, 0.80). **e–h** Nyquist plots of **e** $Li_{6+x}Sn_xAs_{1-x}S_5I$ (x = 0, 0.05, 0.1, 0.6), **f** $Li_{6+x}Sn_xAs_{1-x}S_5I$ (x = 0.15, 0.2, 0.3, 0.4) and **g** $Li_{6+x}Si_xAs_{1-x}S_5I$ (x = 0, 0.1, 0.2, 1), **h** $Li_{6+x}Si_xAs_{1-x}S_5I$ (x = 0.3, 0.4, 0.5, 0.6, 0.7, 0.8, 0.9). Note that the data here are measured in a cell

configuration with stainless steel (SS) as electrode and solid electrolyte (SE) as separator at 25 °C. **i** The ionic conductivity at 25 °C of $Li_{6+x}Sn_xAs_{1-x}S_5I$ and $Li_{6+x}Si_xAs_{1-x}S_5I$ as a function of substitution proportion x. **j** Arrhenius plots of the ionic conductivity of $Li_6AsS_5I$ (LASI), $Li_{6.3}Sn_{0.3}As_{0.7}S_5I$ (LASI-30Sn), $Li_{6.8}Si_{0.8}As_{0.2}S_5I$ (LASI-80Si), and $Li_6PS_5Cl$ (LPSC) as a function of temperature from 0 to 75 °C. **k** The comparison of ionic conductivity at 25 °C and activation energy of representative sulfide SEs reported in previous works and the synthesized $Li_{6.8}Si_{0.8}As_{0.2}S_5I$[32].

enhanced. Notably, an equivalent amount of lithium has been introduced to keep charge conservation when low-valence Sn/Si substitution is conducted. Consequently, it can be speculated that the insufficient Li vacancies to accommodate foreign lithium atoms result in the separation of lithium-contained phases (i.e., $Li_2S$ and LiI).

The solid-solution limit can be determined once the impurity phase containing Sn/Si appears. As shown in Fig. 1a, b, Sn and Si exist separately in the form of $Li_4SnS_4$ and $Li_4SiS_4$, once reaching the solid-solution limit at x = 0.3 and x = 0.8, respectively. After this doping limit, the amount of $Li_4SnS_4$ and $Li_4SiS_4$ further increases until the argyrodite structure collapses, accompanying with the sustained accumulation of lithium-contained phases (i.e., $Li_2S$ and LiI). Interestingly, this solid-solution limit is correlated to the sintering temperature. As shown in Supplementary Figs. 1 and 2, the amount of lithium-contained phases (i.e., $Li_2S$ and LiI) is enhanced after the sintering temperature elevates from 550 °C to 575 °C and 600 °C. More $Li_4SnS_4$ or $Li_4SiS_4$ phase separates when synthesizing LASI-30Sn or LASI-80Si at a higher temperature of 600 °C, which indicates the decrease of solid-solution limit of Sn or Si. In addition, since the ionic radius of $Sn^{4+}$ (69 pm) has a much lower matching degree with that of As (47.5 pm) than $Si^{4+}$ (40 pm) counterpart, it is rational that Sn deserves a much lower solid-solution limit than that of Si. Specifically, the diffraction peaks of Sn-substituted LASI at ~28.91° and 30.20° (corresponding to (311) and (222) crystal planes, respectively) slightly shift to smaller angles (Fig. 1c) as Sn-substitution proportion increases, indicating lattice expansion. On the contrary, diffraction peaks of Si-substituted LASI shift to larger angles (Fig. 1d) as Si-substitution proportion increases, reflecting lattice compression. Further Rietveld refinement results also demonstrate the approximately linear expansion or compression of crystal lattice below the solid-solution limit of Sn or Si, respectively (Supplementary Fig. 3). This lattice parameter/volume variation towards opposite directions is because that the ionic radius of As (47.5 pm) falls between that of $Sn^{4+}$ and $Si^{4+}$. In addition, the relative intensity ratio of these two diffraction peaks at ~28.91° and 30.20° increases with the substitution proportion, as shown in Supplementary Table 2. Without substitution, the intensity of the left diffraction peak is much lower than that of the right one. The relative intensity ratio approaches 1 when the substitution proportions of Sn and Si reach x = 0.1 and x = 0.2, respectively. It further increases as more As is substituted by Si, but slightly decreases when Sn-substitution proportion reaches x = 0.3, which may be caused by the separation of Sn once exceeding the solid-solution limit.

Ionic conductivity is a primary indicator when developing SEs for ASSBs. The ionic conductivities of cold-pressed pellets of $Li_{6+x}Sn_xAs_{1-x}S_5I$ (denoted as LASI-ySn, x = y%, x = 0, 0.05, 0.1, 0.15, 0.2, 0.3, 0.4, 0.6) and $Li_{6+x}Si_xAs_{1-x}S_5I$ (denoted as LASI-ySi, x = y%, x = 0, 0.1, 0.2, 0.3, 0.4, 0.5, 0.6, 0.7, 0.8, 0.9, 1) were investigated via electrochemical impedance spectroscopy (EIS) measurements. As shown in Fig. 1e–h, the Nyquist plots show the typical features of solid superionic conductors, with one semi-circle and a Warburg-like tail[37]. According to the total resistance determined by the intercept on real axis or the intersection between semi-circle and Warburg-like tail, the total ionic conductivity (Supplementary Table 3) can be calculated. The variation of ionic conductivities at 25 °C of LASI-ySn and LASI-ySi as a function of substitution proportion x was plotted in Fig. 1i. These two curves show a volcano-like shape. Specifically, the ionic conductivity rapidly increases with x at a relative low doping concentration, and then smoothly increases until reaching the maximum value. When Sn and Si substitution proportions increase beyond x = 0.3 and x = 0.8, respectively, their ionic conductivities begin to decrease. These two transition points correspond well with their solid-solution limits. Therefore, $Li_{6.3}Sn_{0.3}As_{0.7}S_5I$ (LASI-30Sn) and $Li_{6.8}Si_{0.8}As_{0.2}S_5I$ (LASI-80Si)[32] are selected as the representative materials to investigate the variation of ionic conductivity as a function of temperature within the range of 0–5 °C. Their Arrhenius curves are plotted in Fig. 1j for

calculation of activation energy ($E_a$) based on the Arrhenius equation. Results show that LASI-80Si has higher ionic conductivity and smaller activation energy than those of LASI-30Sn. More importantly, compared with the state-of-the-art argyrodite SE $Li_6PS_5Cl$ (LPSC), LASI-80Si is much better in terms of ionic conductivity at 25 °C and activation energy[32]. To further evaluate the Li-ion conduction property of LASI-80Si in this work, the ionic conductivity and activation energy of representative sulfide SEs reported in previous works are summarized in Fig. 1k and Supplementary Table 4. The highest ionic conductivity for cold-pressed pellets of glass, glass-ceramics and Thio-LISICON (thio-lithium superionic conductor) sulfide SEs (i.e., $Li_2S-P_2S_5-LiI$, $Li_7P_3S_{11}$, and $Li_{3.25}Ge_{0.25}P_{0.75}S_4$) is less than 5 mS cm$^{-1}$ at 25 °C. In comparison, the ionic conductivities of typical sulfides in LGPS family (e.g., $Li_{9.54}Si_{1.74}P_{1.44}S_{11.7}Cl_{0.3}$ (LSPSC)) and argyrodites family (e.g., $Li_{5.5}PS_{4.5}Cl_{1.5}$, $Li_{6.6}Si_{0.6}Sb_{0.4}S_5I$) can reach 10 mS cm$^{-1}$ or even higher at 25 °C. Lithium thioarsenate argyrodites (e.g., $Li_{6.8}Si_{0.8}As_{0.2}S_5I$ (LASI-80Si)) as a distinctive family present a high superionic conductivity (10.4 mS cm$^{-1}$ at 25 °C)[32] on par with those of thiophosphate and thioantimonate counterparts and a competitive activation energy (0.20 eV). In addition, the electronic conductivity of LASI-80Si has also been measured by direct-current polarization and calculated to be $5.03 \times 10^{-9}$ S cm$^{-1}$ at 25 °C, which is six orders of magnitude lower than its ionic conductivity (Supplementary Fig. 4).

## Fast ion conduction mechanism investigations

To further investigate the fast Li$^+$ transport mechanism of Si-substituted LASI, the crystallographic details of LASI-80Si[32] with the highest ionic conductivity are obtained from the Rietveld refinement of X-ray powder diffraction pattern, as shown in Supplementary Fig. 5 and Supplementary Table 5. The unit cell of the thioarsenate phases, $Li_6AsS_5I$, adopts the same cubic $F\bar{4}3m$ space group as $Li_6PS_5I$ (a = 10.1414(1) Å) and $Li_6SbS_5I$ (a = 10.4191(2) Å), but with an intermediate lattice parameter of a = 10.237(1) Å[15]. After Si substitution, the lattice parameter decreases to 10.2100(2) Å, indicating a lattice contraction. Different from the change of lattice parameter for Si-substituted $Li_6SbS_5I$, that of Si-substituted LASI changes mildly due to the relatively smaller radius difference between $Si^{4+}$ (40 pm) and $As^{5+}$ (47.5 pm). It is reported that I$^-$ (2.2 Å) with large radius is difficult to exchange with S$^{2-}$ in $Li_6PS_5I$[38], while the S$^{2-}$/I$^-$ site disorder including 4a (8.4%) and 4c (1.0%) site disorder is activated due to the introduction of Si into LASI. Caused by the large amount of aliovalent substitution of $Si^{4+}$ for $As^{5+}$, extra Li$^+$ ions were inserted into the lattice for charge compensation, which was confirmed by the increased occupancy of Li1 site. However, the interstitial or additional Li sites cannot be accurately identified from Rietveld refinement of XRD data due to the low X-ray form factor of lithium. These obtained structural information of LASI/LASI-80Si[32] are used as the initial structural models and input data for subsequent theoretical calculations/simulations. Moreover, as shown in Supplementary Table 6, the actual chemical composition of LASI-80Si[32] has been quantified to be $Li_{6.835}Si_{0.797}As_{0.202}S_{5.125}I_{1.072}$ via multiple chemical analysis methods, i.e., inductively coupled plasma-atomic emission spectrometry (ICP-AES), carbon and sulfur element analysis and energy dispersive spectrometer (EDS) mapping. The weight fractions of LiI and $Li_2S$ functional phases are quantified to be 2.64 and 2.49 wt% (Supplementary Table 5), respectively.

Ab initio molecular dynamics (AIMD) simulation is generally employed to calculate the ionic conductivity of inorganic SEs. However, the performance of AIMD is usually limited by the system size within hundreds of atoms and time scale within tens of pico-seconds. To extend the simulation time and space scales, molecular dynamics (MD) with deep-learning potential is adopted in this work. The neural network potential models are well trained based on datasets generated by Density Function Theory (DFT) calculation without sacrificing accuracy compared with that of AIMD, which has been specially validated as shown in Supplementary Table 7. Consequently, MD with

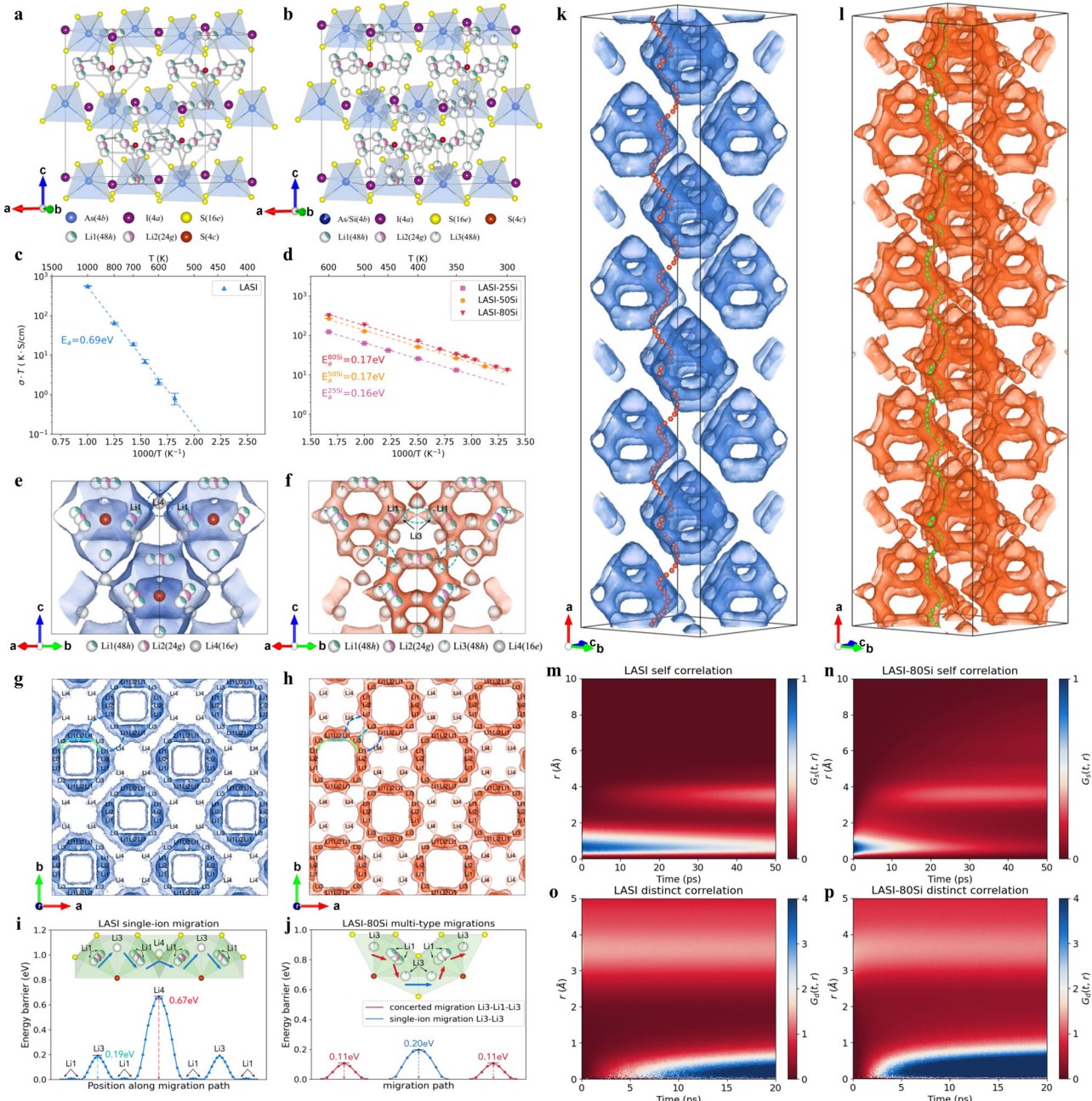

**Fig. 2 | Modeling investigations on the ion conduction mechanism. a, b** Crystal structures of **a** LASI and **b** LASI-80Si after optimization based on first-principle DFT and MD simulation at 400 K. **c, d** Arrhenius plots of the ionic conductivity values as a function of temperatures for **c** LASI, **d** LASI-25Si, LASI-50Si, and LASI-80Si. **e, f** 3D Li-ion probability density distribution of **e** LASI at 1000 K and **f** LASI-80Si at 400 K in the selected three cages of one unit cell. The distinct inter-cage migration pathway for LASI and LASI-80Si is highlighted by the dotted circle. **g, h** 2D Li-ion probability density distribution of **g** LASI at 1000 K and **h** LASI-80Si at 400 K viewed from [001] direction. The doublet jump pathway (Li1(48 h)-Li2(24 g)-Li1(48 h)), intra-cage jump pathway (Li1(48 h)-Li3(48 h)-Li1(48 h), inter-cage jump pathway (Li1(48 h)-Li4(16e)-Li1(48 h) and new inter-cage jump pathway (Li3(48 h)-

Li3(48 h)) are represented by the bidirectional arrows with light blue, light green, dark blue and dark green color, respectively, in the top left corner of (**g**) and (**h**). **i, j** Migration energy barrier in **i** single-ion migration mode for LASI and **j** multi-type migration modes for LASI-80Si. The corresponding Li-ion migration pathways along channel across multiple Li sites are shown in insets. **k, l** The 3D Li-ion probability density distribution in the 4 × 1 × 1 supercell of **k** LASI and **l** LASI-80Si at the same isosurface level. The Li-ion trajectories along a-axis is used for visual guidance. **m–p** Self van Hove correlation function of **m** LASI and **n** LASI-80Si, and distinct van Hove correlation function of **o** LASI and **p** LASI-80Si during MD simulations at 600 K.

deep-learning potential runs 100–1000 times faster than AIMD while system size is also ten times larger than that of AIMD. The details of computational methods, model training, MD simulation settings (Supplementary Tables 8 and 9) and software packages[39–42] used in this work are summarized in Supplementary Materials.

As shown in Fig. 2a, b, the crystal structures of LASI and LASI-80Si[32] are first obtained after optimization based on first-principle DFT and

MD simulation at 400 K. Iodine anions constitute the face-centered cubic host lattice with [As/SiS$_4$] tetrahedrons occupying the octahedral voids and [Li$_6$S] pseudo-octahedral cages residing in half of the tetrahedral voids[36]. While only two types of Li sites (i.e., Li1(48 h) and Li2(24 g)) exist for pristine LASI, two novel Li sites (i.e., Li3(48 h) and Li4(16e)) are generated after introducing additional Li$^+$ into structure by aliovalent substitution. This is confirmed by the enhanced spatial

occupancy (Supplementary Tables 10 and 11) of Li3 and Li4 sites for LASI-80Si[32]. The ionic conductivities at different temperatures for LASI (Fig. 2c and Supplementary Table 12) and LASI-xSi (Fig. 2d and Supplementary Tables 13–15) are obtained from MD simulation using deep-learning potential. The MD simulation of LASI was performed above 500 K considering its low ionic conductivity and high activation energy (0.69 eV, Supplementary Table 16). Based on Arrhenius equation, the ionic conductivity at 25 °C (ca. $1 \times 10^{-5}$ mS cm$^{-1}$) of LASI can be deduced through extrapolation. In stark contrast, LASI-80Si[32] possesses a high ionic conductivity (44.5 mS cm$^{-1}$ at 300 K) and low activation energy (0.17 eV, Supplementary Table 16). It is noteworthy that such enhancing effect of Si substitution on ionic conductivity corresponds well with the aforementioned experimental results. Moreover, the influence of Li site occupancy and probability density distribution before and after Si substitution on Li-ion transportation mechanism and activation energy was further investigated. Figures 2e, f specifically demonstrate Li sites and Li-ion probability density of LASI and LASI-80Si[32] in the selected three cages within one unit cell. While LASI has only one inter-cage migration path (Li1(48 h)-Li4(16e)-Li1(48 h)) highlighted by the blue dotted circle in Fig. 2e, LASI-80Si[32] presents an additional dominant inter-cage migration path (Li3(48 h)-Li3(48 h), green dotted circle in Fig. 2f). Due to the high degree of symmetry for cubic structure in space group $F\bar{4}3m$, the mean square displacement (MSD) of LASI and LASI-80Si[32] (Supplementary Fig. 6) is equivalent for arbitrary axial directions and the other properties (e.g., ionic conductivity, Li-ion probability density) are also isotropic. Herein, (001) crystalline plane along c-axis was chosen to display the 2D (two-dimensional) Li-ion probability density distribution, which was obtained from MD simulations of LASI and LASI-80Si[32] (Fig. 2g, h, and Supplementary Fig. 7). The color brightness of the iso-surfaces represents different Li-ion densities, where the darker color means a higher Li-ion density. Four types of Li-ion migration pathways are identified and represented by the bidirectional arrows with different colors in the top left corner of Fig. 2g, h. Doublet jump pathway (Li1(48 h)-Li2(24 g)-Li1(48 h)) between adjacent Li1 pair is represented by the light blue arrow. It is worth noting that a new Li3(48 h) site locates between two neighboring unpaired Li1(48 h) sites along the intra-cage jump pathway (Li1(48 h)-Li1(48 h), light green arrow) for LASI-80Si[32] compared to that of LASI. Inter-cage jump pathway (Li1(48 h)-Li4(16e)-Li1(48 h), dark blue arrow) exists for both LASI and LASI-80Si[32]. However, the new inter-cage jump pathway (Li3(48 h)-Li3(48 h), dark green arrow) bridged by two Li3(48 h) sites in two adjacent cages is only observed in LASI-80Si[32]. As the total Li-ion occupancy ($Occ(Li1) \times 2 + Occ(Li2)$) in Li1(48 h)-Li2(24 g)-Li1(48 h) pair equals 1.0 for both LASI (Supplementary Table 10) and LASI-80Si[32] (Supplementary Table 11), it is speculated that low-energy sites are fully occupied. The additional Li$^+$ ions introduced by aliovalent substitution prefer to occupy high-energy sites (i.e., Li3 and Li4) given the Coulomb interactions among these mobile ions, which can be confirmed by the chubby iso-surfaces around Li3 and Li4 for LASI-80Si[32] compared to those of LASI.

Figure 2i shows the energy landscapes as a function of position along the single-ion migration pathway of LASI. While the intra-cage migration exhibits a relatively low energy barrier of 0.19 eV, the inter-cage migration shows a large energy barrier of 0.67 eV. This is consistent with the activation energy (0.69 eV) calculated by MD simulation, since inter-cage migration is dominant for long-range diffusion. The energy barrier of LASI-80Si[32] (Fig. 2j) is individually calculated by nudged elastic band (NEB) method for intra-cage concerted migration and inter-cage single-ion migration, which exhibit a lower energy barrier of 0.11 eV and 0.20 eV, respectively. Consequently, the occupancy enhancement at Li3 sites facilitates both intra-cage and inter-cage migration. On one hand, the occupied high-energy Li3 site along the original intra-cage pathway (Li1-Li1) activates the concerted migration[43,44] fashion (Li3-Li1-Li3) with low activation energy, which has been directly demonstrated by the time-series trajectories obtained

from MD simulation (Supplementary Fig. 8). On the other hand, the new inter-cage jump pathway bridged by two Li3(48 h) sites in two adjacent cages also significantly lower the energy barrier from 0.67 eV to 0.20 eV. Specifically, the Li-ion migration trajectories along a-axis in $4 \times 1 \times 1$ supercell with lowest global activation energy for LASI and LASI-80Si[32] is stereoscopically depicted in Fig. 2k, l, respectively. LASI-80Si[32] presents a well-connected Li-ion percolation network compared to that of LASI, especially for inter-cage migration. The distinct intra-cage and inter-cage migration pathways for LASI and LASI-80Si[32] are further manifested in 3D (three-dimensional) scenario. The time correlation of Li$^+$ hopping during migration was further investigated by van Hove time-space correlation function for both LASI and LASI-80Si[32] (Fig. 2m–p) at 600 K. LASI (Fig. 2m) shows a strong self-correlation within 1 Å over 50 ps compared to LASI-80Si[32] (Fig. 2n) on account of the long-term distribution of the dark blue zone, which suggests the sluggish Li$^+$ hopping (or high energy barrier) between two low-energy Li1 sites. The darker red color in the top right zone (above 5 Å) of Fig. 2m indicates that the inter-cage jumps of Li$^+$ in LASI are less frequent than that of LASI-80Si[32] (Fig. 2n) at 600 K. In terms of distinct correlation (Fig. 2o, p), LASI-80Si[32] presents a strong peak near 0 Å and 2.5 ps (Fig. 2p), which indicates the higher probability to observe one Li ion rapidly occupying the vacant site left behind by another one within a few pico-seconds compared to that of LASI (Fig. 2o), corroborating again the concerted migration activated by Si substitution.

## Understanding the stability in humid air environment and the electrochemical properties of sulfide solid electrolytes

Air stability is a critical parameter to evaluate the practicability of sulfide SEs. The state-of-the-art phosphorus-based sulfide SEs suffer from poor air stability and strong hygroscopicity, generating toxic H$_2$S gas with damaged structure/performance in air[17,19,45]. Herein, the H$_2$S generation amount/rate, structural evolution, and hydrolysis reaction products of LASI-80Si and typical sulfide SEs (LPSI, LPSC, and LSPSC) were systematically investigated by exposing these samples to humid air. The H$_2$S generation amount was measured by our specially-designed detection system (Supplementary Fig. 9)[25], with a relative humidity (RH) of 23–25% and a gas flow rate of 250 cm$^3$ min$^{-1}$. As shown in Fig. 3a, the normalized total H$_2$S generation amount of argyrodite-type sulfide SEs (LPSI, LPSC, LASI, and LASI-80Si) instantly increases and then reaches the maximum value after moisture exposure for ~20 min. In spite of this similar variation trend, their air stabilities can still be compared with each other by means of the total generation amount of H$_2$S. LASI and LASI-80Si show better air stability than phosphorus-based argyrodites (LPSI and LPSC), given that LPSI and LPSC release much larger amount of H$_2$S gas (105.35 cm$^3$ g$^{-1}$ and 98.99 cm$^3$ g$^{-1}$, respectively, Supplementary Table 17). In contrast, LASI and LASI-80Si generate 75.07 cm$^3$ g$^{-1}$ and 91.32 cm$^3$ g$^{-1}$ H$_2$S gas under the same exposure condition. Therefore, the air stability of these four argyrodite-type sulfide SEs follows the order of LASI > LASI-80Si > LPSC > LPSI. This is consistent with the HSAB theory and the thermodynamic simulation results postulated by Mo et al.[20,29,46], based on which the moisture stability of sulfide SEs containing these central cations follows the trend of As$^{5+}$ > Si$^{4+}$ > P$^{5+}$. Although Si substitution improves the ionic conductivity of LASI, it adversely affects the air stability in a certain degree. Figure 3b shows the generation rate of H$_2$S obtained by calculating the first-order derivative of curves in Fig. 3a. The generation rate of H$_2$S for LPSI reaches the peak value more quickly ($t = 0.3333$ min) but slowly drops to zero, very different from that of the other three sulfide SEs, as shown in Fig. 3b and Supplementary Table 17. While the peak positions of H$_2$S generation rate for LPSC, LASI, and LASI-80Si all locate at $t = 0.4167$ min, the peak values for LASI (37.19 cm$^3$ g$^{-1}$ min$^{-1}$) and LASI-80Si (39.89 cm$^3$ g$^{-1}$ min$^{-1}$) are relatively smaller than that of LPSC (49.95 cm$^3$ g$^{-1}$ min$^{-1}$). Therefore, the air stability of these four sulfide SEs can be concluded to follow the order of LASI > LASI-

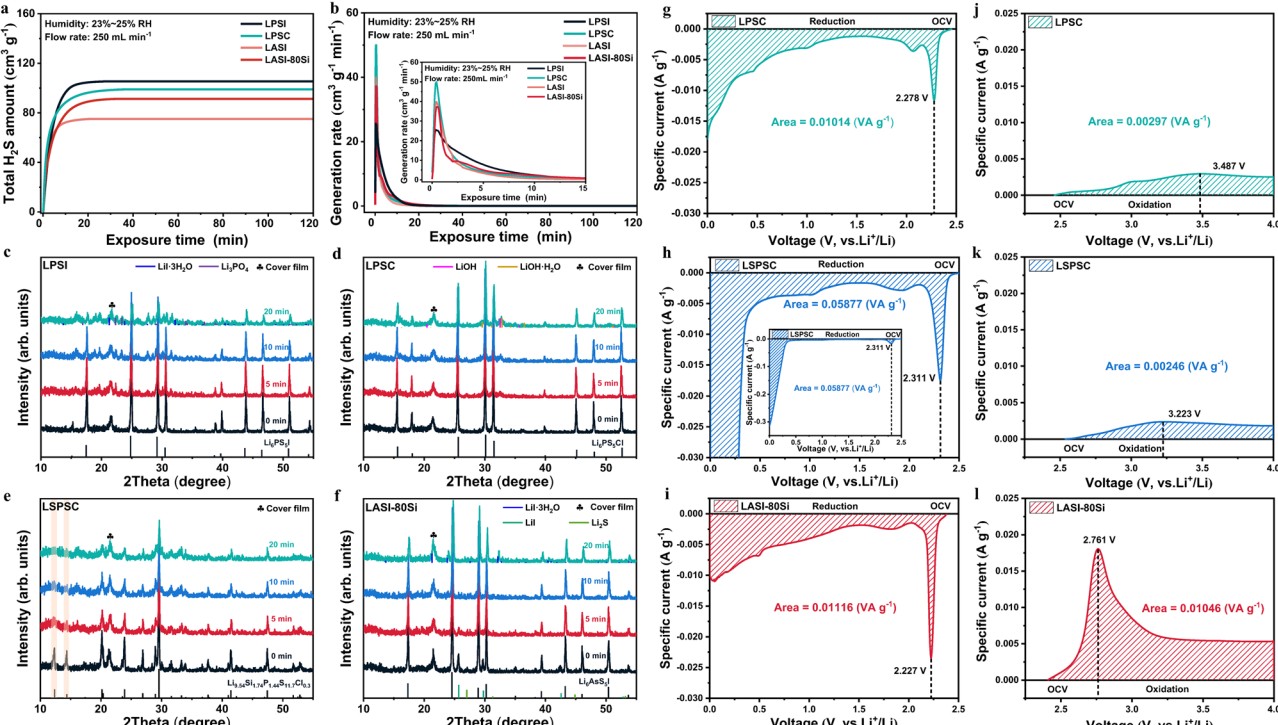

**Fig. 3 | Stability in humid air environment and electrochemical properties of sulfide solid electrolytes. a** The total generation amount of H₂S of LPSI, LPSC, LASI, and LASI-80Si when exposed to flowing air with a flow rate of 250 mL min⁻¹ and relative humidity of 23–25% RH. **b** The generation rate of H₂S obtained by calculating the first-order derivative of curves in (**a**). **c**–**f** XRD patterns of **c** LPSI, **d** LPSC, **e** LSPSC and **f** LASI-80Si after exposure to humid air (23–25% RH) for 0, 5, 10, and 20 min. **g**–**l** Linear sweep voltammetry profiles of Li|SE|SE/C cells tested at 25 °C to evaluate the electrochemical stability of **g** LPSC, **h** LSPSC, and **i** LASI-80Si in reduction at low voltage (OCV–0.0 V) and **j** LPSC, **k** LSPSC, and **l** LASI-80Si in oxidation at high voltage (OCV–4.0 V).

80Si > LPSI > LPSC, in terms of the total amount and generation rate of H₂S gas.

To make a comprehensive and quantitative comparison, structural retention rate is defined and obtained as the intensity ratio between the strongest diffraction peak after and before exposure to humid air. As shown in Fig. 3c, some diffraction peaks (e.g., 2θ = 15.28°) of LPSI weakened or vanished, and impurity peaks from side products (e.g., LiI·3H₂O and Li₃PO₄) appeared, indicating the rapid structural degradation of LPSI even at low humidity (23–25% RH). The structural retention rate (Supplementary Table 18) rapidly drops from 63.93% to 17.04% as exposure time increases from 5 min to 20 min. In contrast, all diffraction peaks of LPSC (Fig. 3d) keep almost unchanged, despite some small diffraction peaks from side products of LiOH and LiOH·H₂O, which may be induced by Li⁺ loss due to Li⁺/H⁺ exchange on LPSC surface[47]. Compared with argyrodite-type sulfide SEs, LSPSC in LGPS family undergoes much more sever amorphization (Fig. 3e) with declined intensity of the strongest diffraction peak at 2θ = 29.56° to 39.98% and broadening of diffraction peaks (e.g., 2θ = 12.32°, 14.36°). The hydrolysis products of LSPSC may exist as amorphous phase, which cannot be identified from the broad impurity peaks (e.g., 2θ ~ 16°). Interestingly, while all diffraction peaks of LASI-80Si remain almost unchanged (Fig. 3f), the low-content LiI and Li₂S functional phases react with H₂O molecules to generate LiI·3H₂O hydrate and some unidentifiable amorphous phases. In addition to the high air stability of LASI-80Si, the competitive reactions with H₂O molecules between LASI-80Si and its functional phases (LiI and Li₂S) also enable the highest structure retention of 91.34% after exposure for 5 min (Supplementary Table 18). The little change on crystalline structure of LASI-80Si further corroborates its improved air stability than the iodine-based analog LPSI. It is worth noting that a similar phenomenon has also been observed for Li₄PS₄I[48], in which LiI suppressed the generation of H₂S through the formation of LiI·H₂O hydrate. Therefore, the

air stability of these sulfide SEs follows the order of LASI-80Si > LPSC > LSPSC > LPSI, in terms of the structure retention after exposure for 5 min. After a comprehensive comparison of these sulfide SEs on the generation amount/rate of H₂S gas, structure retention, and hydrolysis products, it can be concluded that LASI-80Si possesses a much better air stability than state-of-the-art LPSI/LPSC argyrodites and LSPSC in LGPS family.

The electrochemical stabilities of LPSC, LSPSC and LASI-80Si were characterized by linear sweep voltammetry (LSV) at a scan rate of 0.1 mV/s to determine the proper working voltage range and active electrode materials suitable for ASSBs. As shown in Fig. 3g–l, all three sulfide SEs start to be reduced/oxidized at around open-circuit voltage (OCV). Thus, it is advisable that the peak position and integral specific current[49] instead of electrochemical stability window (ESW) should be selected as the indicators to evaluate the electrochemical stability of these three sulfide SEs. As summarized in Supplementary Table 19, the first reduction peak of LPSC, LSPSC, and LASI-80Si locates at 2.278 V, 2.311 V, and 2.227 V, respectively. The integral reduction specific current of LPSC, LSPSC, and LASI-80Si are calculated to be 0.01014 VA g⁻¹, 0.05877 VA g⁻¹, and 0.01116 VA g⁻¹, respectively. While LSPSC exhibits poor reduction stability given the higher reduction voltage and integral reduction specific current than those of LPSC and LASI-80Si, the reduction stabilities of LPSC and LASI-80Si are similar. The overall reduction stability of these sulfide SEs follows the order of LPSC > LASI-80Si > LSPSC. The first oxidation peak of LPSC, LSPSC, and LASI-80Si locates at 3.487 V, 3.223 V, and 2.761 V, respectively. The integral oxidation specific current of LPSC, LSPSC, and LASI-80Si are calculated to be 0.00297 VA g⁻¹, 0.00246 VA g⁻¹, and 0.01046 VA g⁻¹, respectively. It is noteworthy that LASI-80Si is more vulnerable to oxidation decomposition due to its lower oxidation peak (<3.0 V) and higher integral oxidation-specific current compared to those of LPSC and LSPSC. Therefore, the oxidation stability of these sulfide SEs follows the order

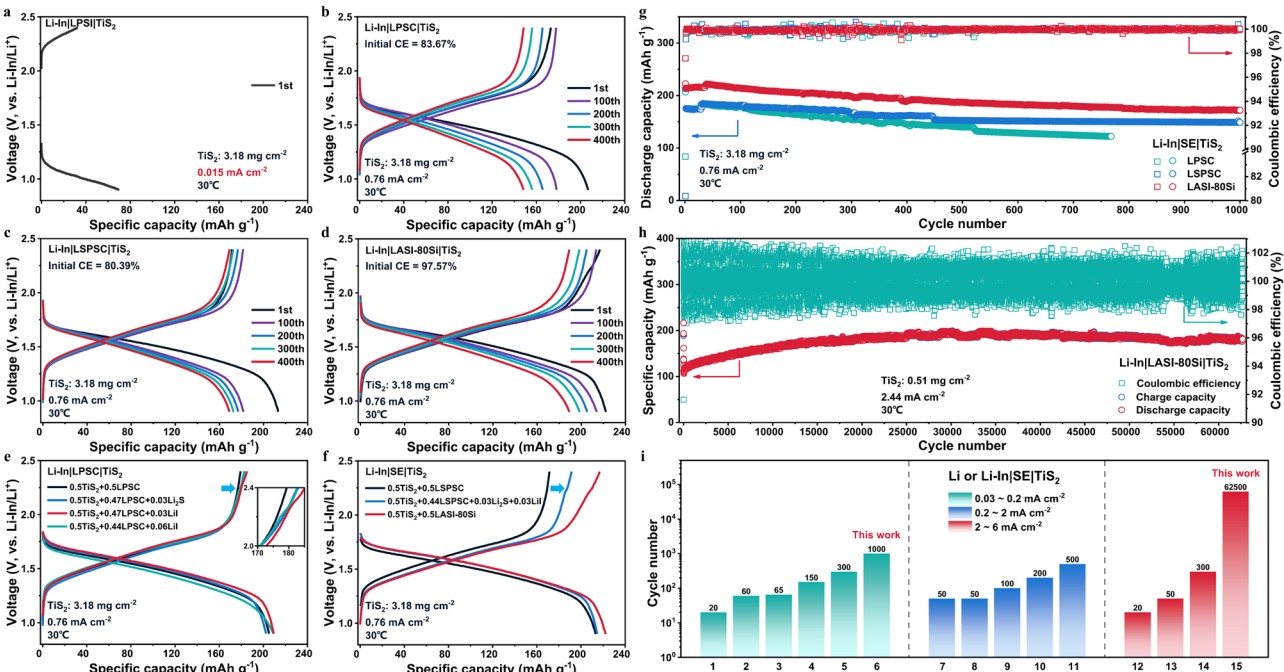

**Fig. 4 | Charge–discharge profiles and long-term cyclability.** The charge–discharge profiles of Li-In|SE|TiS$_2$ASSBs with **a** LPSI, **b** LPSC, **c** LSPSC, and **d** LASI-80Si sulfide SEs, respectively. Due to the low ionic conductivity of LPSI (~$10^{-6}$ S cm$^{-1}$ at 25 °C) compared with the other three sulfide SEs, Li-In|LPSI|TiS$_2$ASSB can only cycle at low current density of 0.015 mA cm$^{-2}$ and delivers a low specific capacity. **e** Comparison of 1$^{st}$-cycle charge–discharge profiles for TiS$_2$ composite cathodes without and with Li$_2$S/LiI additives, including 0.5TiS$_2$ + 0.5LPSC (black), 0.5TiS$_2$ + 0.47LPSC + 0.03Li$_2$S (blue), 0.5TiS$_2$ + 0.47LPSC + 0.03LiI (red), and 0.5TiS$_2$ + 0.47LPSC + 0.06LiI (cyan). **f** Comparison of 1$^{st}$-cycle charge–discharge profiles for various TiS$_2$ composite cathodes, including 0.5TiS$_2$ + 0.5LSPSC (black), 0.5TiS$_2$ + 0.44LSPSC + 0.03Li$_2$S + 0.03LiI (blue), and 0.5TiS$_2$ + 0.5LASI-80Si (red). The numerical value before each component represents its weight ratio in composite cathodes. **g** The discharge capacity and coulombic efficiency as a function of cycle number at 0.76 mA cm$^{-2}$ for ASSBs with LPSC, LSPSC, and LASI-80Si sulfide SEs, respectively. **h** The charge/discharge capacity and coulombic efficiency as a function of cycle number at 2.44 mA cm$^{-2}$ for Li-In|LASI-80Si|TiS$_2$ ASSB. The ASSB first cycled at 0.122 and 1.22 mA cm$^{-2}$ for five cycles, and subsequently cycled at 2.44 mA cm$^{-2}$. **i** Comparison of cycle lives at different current densities for TiS$_2$-based ASSBs coupling with various electrolytes, including the borohydride SEs[58–62], glass[63], glass-ceramics[64], Thio-LISICON[27], LGPS-type[65,66], and argyrodite-type[67] sulfide SEs, and LASI-80Si in this work. Note that the number at x-axis corresponds to the No. at Supplementary Table 24. All cells assembled in this work were tested at 30 °C.

of LSPSC > LPSC > LASI-80Si. Despite the improved ionic conductivity of LASI-80Si, its poor oxidation stability compared with LPSC and LSPSC may become a restriction for its application in ASSBs. These LSV results suggest that metal sulfide cathode with a working voltage in the range of 1–3 V (vs. Li$^+$/Li) may be more suitable for these sulfide SEs than conventional high-voltage oxide cathodes[50]. Specifically, when matching with Li$_2$ZrO$_3$-coated Ni-rich cathode (i.e., LiNi$_{0.90}$Mn$_{0.05}$Co$_{0.05}$O$_2$), LASI-80Si ASSB exhibits low initial coulombic efficiency and rapid capacity degradation (Supplementary Figs. 10 and 11), which indicates the unstable cyclability may originate from the low oxidation stability of the iodine-based argyrodite materials.

**Electrochemical energy storage performance of sulfide solid-state electrolytes in Li-In||TiS$_2$ cells**
To evaluate the electrochemical performances of aforementioned sulfide SEs, ASSBs with metal sulfide cathode, Li-In alloy anode, and LPSI, LPSC, LSPSC, or LASI-80Si SE were assembled. Since conversion-type cathodes (e.g., FeS$_2$)[32] suffer from large volume change, sluggish kinetics, and short life span, ASSBs coupling LASI-80Si SE with the intercalation-type TiS$_2$ cathode are specially investigated in this work. The intercalation reaction of TiS$_2$ at 0.9-2.4 V (vs. Li-In/Li$^+$) follows Eq. (1):

$$TiS_2 + xLi^+ + xe^- \leftrightarrow Li_xTiS_2 \ (0 \le x \le 1) \tag{1}$$

As shown in Fig. 4a, TiS$_2$|LPSI|Li-In ASSB can only work at a low current density of 0.015 mA cm$^{-2}$ and presents a low specific capacity, due to the small ionic conductivity of $3.33 \times 10^{-6}$ S cm$^{-1}$ at 25 °C for LPSI,

which restricts the reaction kinetics and induces a large overpotential. Therefore, high ionic conductivity is a prerequisite for SEs to ensure effective cycling and high power of ASSBs. ASSBs using the other three sulfide SEs with high ionic conductivity (≥$10^{-3}$ S cm$^{-1}$ at 25 °C) exhibit specific discharge capacities over 220 mAh g$^{-1}$ at a current density of 0.076 mA cm$^{-2}$ (Supplementary Fig. 12 and Supplementary Table 20), close to the theoretical capacity of TiS$_2$ cathode (239 mAh g$^{-1}$). It is worth noting that the initial coulombic efficiency (CE) of ASSB with LASI-80Si is even larger than 100% (Supplementary Table 20), distinctly different from that of LPSC and LSPSC (94.48% and 92.67%, respectively). In addition, the first-cycle charge curve (Supplementary Fig. 12) of ASSB with LASI-80Si is bended at ~2.16 V (vs. Li-In/Li$^+$)[51], which corresponds to the oxidation peak at 2.761 V (vs. Li$^+$/Li, Fig. 3l) considering the voltage plateau of Li-In alloy (0.6 V vs. Li$^+$/Li). Therefore, the oxidation of LASI-80Si during charging may lead to a voltage profile bending at ~2.16 V and the abnormal initial CE larger than 100% at 0.076 mA cm$^{-2}$ (Supplementary Fig. 12). In contrast, the first-cycle charge curve (Supplementary Fig. 7) and initial CE (Supplementary Table 20) of LPSC and LSPSC do not show anomalous behavior, due to their milder decomposition reactions within the working voltage range of TiS$_2$. When increasing the current density to 0.76 mA cm$^{-2}$, the specific charge capacity and initial CE (Supplementary Table 21) of ASSBs with LPSC and LSPSC decrease from ~210 mAh g$^{-1}$ and >90% to >170 mAh g$^{-1}$ and >80%, respectively. However, a large irreversible capacity exists for LPSC/LSPSC ASSBs at a higher current density of 0.76 mA cm$^{-2}$, as shown in Fig. 4b, c. In stark contrast, the specific discharge/charge capacity and initial CE of LASI-80Si ASSB are higher, with a drop from 227.9 mAh g$^{-1}$/228.7 mAh g$^{-1}$ and 100.33% to 222.3

mAh g$^{-1}$/216.9 mAh g$^{-1}$ and 97.57%, respectively (Supplementary Tables 20 and 21). In addition, while the initial charge curves of LPSC/LSPSC ASSBs present a steep slope above 2.0 V (Fig. 4b, c), that of LASI-80Si ASSB displays a gentle slope above 2.0 V and a bending at ~2.16 V (Fig. 4d). Interestingly, the bending disappears for the subsequent charge curves of LASI-80Si ASSB, indicating that an interfacial passivation effect has been trigged. Similar phenomena have also been observed for the first-cycle charge profile of ASSBs with $30Li_2S\cdot25B_2S_3\cdot45LiI\cdot25SiO_2$ and $Li_{6.7}Si_{0.7}Sb_{0.3}S_5I$[15,52], which are ascribed to SE oxidation at the cathode interface during initial cycling, given the poor electrochemical stability of sulfide SEs. However, the possible delithiation of LiI impurity during charging, which has been identified from NMR spectra or XRD patterns of $30Li_2S\cdot25B_2S_3\cdot45LiI\cdot25SiO_2$ and $Li_{6.7}Si_{0.7}Sb_{0.3}S_5I$, seems to be excluded or neglected.

To further investigate the underlying mechanism for high (dis)charge capacity of LASI-80Si ASSB, X-ray photoelectron spectroscopy (XPS) characterization on the interface between $TiS_2$ cathode and LASI-80Si or LSPSC SE before and after cycling was performed. The chemical species of $TiS_2$/LASI-80Si composite electrode before and after cycling are found to be similar with those of $TiS_2$/LSPSC counterpart, as shown in Supplementary Figs. 13 and 14. No evidence shows continuous electrochemical decomposition of LASI-80Si or interfacial reactions between LASI-80Si and $TiS_2$. Interestingly, $LiTiS_2$ content on the surface of $TiS_2$/LASI-80Si composite electrode is much less than that of $TiS_2$/LSPSC counterpart based on the relative intensity ratio between $LiTiS_2$ and $TiS_2$ (from S 2p and Ti 2p spectra, Supplementary Figs. 13 and 14) under full-charge state, suggesting the improved delithiation kinetics of $TiS_2$/LASI-80Si composite electrode. Besides, the relatively stronger intensity of $Li_2S$ for $TiS_2$/LASI-80Si composite electrode than that of $TiS_2$/LSPSC counterpart after cycling indicates a slight electrochemical decomposition of LASI-80Si into $Li_2S$ and LiI products. Given the high theoretical capacity of $Li_2S$ (1166 mAh g$^{-1}$) and LiI (200 mAh g$^{-1}$), and the catalytic effect[53,54] of $TiS_2$ and LiI on $Li_2S$/S conversion reaction, $Li_2S$ and LiI functional phases are speculated to bring about the improved specific charge capacity, initial CE and rate capability. To prove this assumption, 3 wt% $Li_2S$ or LiI, a similar content in as-synthesized LASI-80Si, was first introduced during the preparation of $TiS_2$/LPSC composite electrode. Both $Li_2S$- and LiI-incorporated composite cathodes exhibit an enhanced specific charge capacity (Fig. 4e), initial CE (Supplementary Table 22), and rate capability (Supplementary Fig. 15). It is worth noting that the mass of $Li_2S$ or LiI additive is not taken into account when calculating the specific capacity for fair comparison. Due to the effective conduction networks constructed by $TiS_2$ and LASI-80Si and the catalytic effect of $TiS_2$, the introduction of a small amount of $Li_2S$ or LiI has negligible adverse effect on the electrochemical performances. However, the rate capability deteriorated after increasing the LiI content to 6 wt%, under which Li$^+$ transportation may be impeded (as shown in Supplementary Fig. 15). In addition, it is interesting that the introduction of LiI induces the bending during the first-cycle charge, and that of $Li_2S$ results in a gentle slope which may correspond to the delithiation process of LiI and $Li_2S$, respectively (amplified inset in Fig. 4e). To further prove this point, both $Li_2S$ (3 wt%) and LiI (3 wt%) were added into $TiS_2$/LSPSC composite electrode, maintaining the constant $TiS_2$ content of 50% (Fig. 4f). In addition to the increased initial CE from 80.39% to 89.19% (Supplementary Table 23), the specific charge capacity and rate capability have also been improved (Supplementary Fig. 16). Nevertheless, despite the similar ionic conductivity between LSPSC and LASI-80Si, the boosting effect of $Li_2S$ and LiI additives into $TiS_2$/LSPSC composite cathode is still lower to that of $TiS_2$/LASI-80Si composite cathode (Fig. 4f), in which $Li_2S$ and LiI may distribute more uniformly on the surface of LASI-80Si particle, as a result of their in situ formation/separation during the preparation of LASI-80Si sulfide SE. Moreover, the possible complex defects generated in $Li_2S$ crystals after a series of treatment (ball milling, heat treatment and interaction with LiI) may

improve the conductivity of $Li_2S$[55]. The intimate contact among $TiS_2$, LiI, and $Li_2S$ may also facilitate the catalytic effect of $TiS_2$ and LiI on lowering the barrier of $Li_2S$/S conversion, thus producing more capacity. Similar phenomena including the bending at ~2.2 V (vs. Li-In/Li$^+$) and a gentle slope of the first-cycle charging curve were also observed for $Li_2S$ and LiI incorporated $TiS_2$/LSPSC composite cathode. Therefore, it is reasonable to conclude that the in situ formed and uniformly distributed $Li_2S$ and LiI functional phases in LASI-80Si serve as lithium supplements by delithiation to replenish Li$^+$ sacrificed in interphase formation or trapped in $TiS_2$ interlayer during charging. This contributes to the large specific charge capacity (228.7 mAh g$^{-1}$ at 0.076 mA cm$^{-2}$ and 216.9 mAh g$^{-1}$ at 0.76 mA cm$^{-2}$), high initial CE (100.33% at 0.076 mA cm$^{-2}$ and 97.57% at 0.76 mA cm$^{-2}$), and excellent rate capability of LASI-80Si ASSB.

The long-term cycle performance of ASSBs with LPSC, LSPSC, and LASI-80Si at 0.76 mA cm$^{-2}$ is shown in Fig. 4g. While LPSC/LSPSC ASSBs show a similar initial charge capacity of >170 mAh g$^{-1}$ (Supplementary Table 21), ASSB using LSPSC with a superionic conductivity of 10.1 mS cm$^{-1}$ at 25 °C exhibits a much higher capacity retention of 84.88% after 1000 cycles than that of LPSC (69.67% after 768 cycles). In contrast, LASI-80Si ASSB possesses a much larger initial charge capacity of 216.9 mAh g$^{-1}$, and a moderate capacity retention of 80.26% after 1000 cycles. Due to the discrepancy in initial CE for ASSBs with different sulfide SEs, we arbitrarily decided to calculate the capacity retention rate by using the second-cycle specific discharge capacity rather than the first one. After 1000 cycles, LASI-80Si ASSB still exhibits a much larger specific discharge capacity of 172 mAh g$^{-1}$ than LSPSC ASSB (148.8 mAh g$^{-1}$). More importantly, even at 2.44 mA cm$^{-2}$, LASI-80Si ASSB still demonstrates excellent long-term cycle stability (62,500 cycles) and high specific discharge capacity, as shown in Fig. 4h. Although an initial activation process under low current densities (0.12 mA cm$^{-2}$ and 1.22 mA cm$^{-2}$ for 5 cycles, respectively) has been performed, its specific capacity gradually increases from 112.7 to 198.5 mAh g$^{-1}$ with cycle number. Sequentially, the specific capacity retains around ~180 mAh g$^{-1}$ with slight fluctuation and degradation. This cycle stability is associated by standard discharge-charge profiles (Supplementary Fig. 17) and steady fluctuation of CE at high current rate, with the possibility of short circuit aroused by Li-In dendrite growth[56] cautiously excluded. This large capacity increase at initial stage may be caused by the self-heating healing mechanism[57] that can only be triggered under high current densities. Joule heating under high current density can not only fuse dendrites to elongate cycle life, but also facilitate Li$^+$ diffusion and promote reaction kinetics for gradually improved capacity. To further evaluate the long-term cycle stability of LASI-80Si ASSBs with $TiS_2$ cathode, a systematic comparison of cycle lives at different current densities has been made (as summarized in Fig. 4i and Supplementary Table 24) for $TiS_2$ ASSBs with various SEs, including borohydride SEs[58–62], glass[63,64], glass-ceramics[64], Thio-LISICON[27], LGPS-type[65,66], and argyrodite-type[67] sulfide SEs. Compared with the state-of-the-art performance of $Li_6PS_5Cl$ ASSB (500 cycles at 1.95 mA cm$^{-2}$)[67], the LASI-80Si ASSBs in this work demonstrate an excellent long-term cycle stability with cycle number of 1000 cycles and 62,500 cycles at 0.12 mA cm$^{-2}$ and 2.44 mA cm$^{-2}$, respectively. These experimental results are well-positioned compared to the state-of-the-art of $TiS_2$-based ASSBs (Supplementary Table 24).

Rate capability, as a crucial indicator, to achieve higher power density. Although sulfide SEs can provide superionic conductivity, the development of high-power ASSBs is still hindered by the sluggish transport of Li$^+$ and e$^-$ caused by structural stability of cathode active materials, interfacial compatibility between electrolyte and electrode, and effectiveness of conductive network. Figure 5a shows the rate capability of ASSBs with LPSC, LSPSC, and LASI-80Si, by gradually increasing the current density from 0.76 to 1.52, 2.28, 3.04, 3.8, 7.6 mA cm$^{-2}$ and eventually returning back to 0.76 mA cm$^{-2}$. As summarized in Supplementary Table 25, the discharge capacities of LPSC

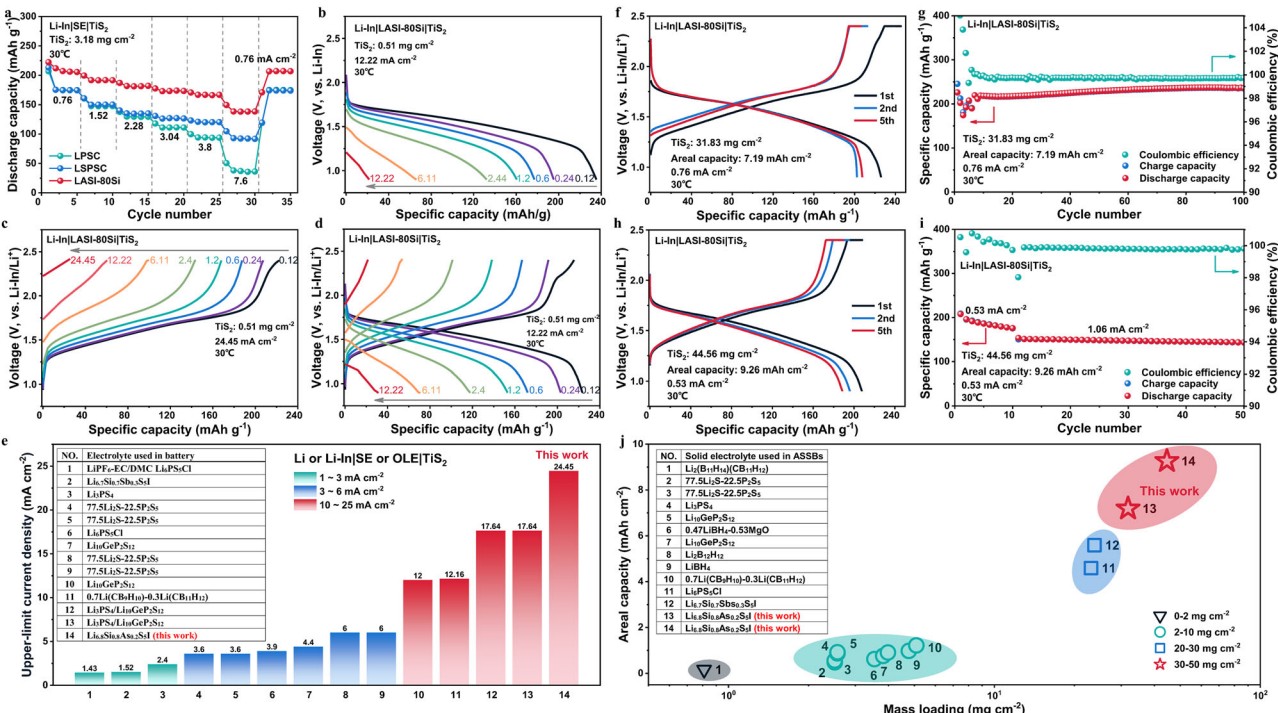

**Fig. 5 | Rate capability and high-mass-loading electrode performances. a** The discharge capacity as a function of cycle number at different current densities for Li-In|SE|TiS₂ASSBs with LPSC, LSPSC, and LASI-80Si sulfide SEs. The charge–discharge profiles of ASSBs with LASI-80Si under different charge–discharge protocols, including **b** discharging at an increased current density from 0.12 to 12.22 mA cm⁻² but charging at a constant current density of 0.12 mA cm⁻², **c** discharging at a constant current density of 0.12 mA cm⁻² but charging at an increased current density from 0.12 to 24.45 cm⁻² and **d** discharging and charging at the same increased current density from 0.12 to 12.22 mA cm⁻². **e** The comparison of upper-limit current density for TiS₂-based batteries with organic liquid electrolyte (OLE)[79], borohydride SEs[58], glass[63], glass-ceramics[70], Thio-LISICON[27], LGPS-type[65,66,70], and argyrodite-type[15,67] sulfide SEs and LASI-80Si in this

work. **f** The charge–discharge profiles of Li-In|LASI-80Si|TiS₂ASSB with a high mass loading of 31.83 mg cm⁻² at 0.76 mA cm⁻². **g** The charge and discharge capacity and coulombic efficiency λas a function of cycle number at 0.76 mA cm⁻² for Li-In|LASI-80Si|TiS₂ ASSB. **h** The charge–discharge profiles of TiS₂|LASI-80Si | Li-In ASSB with a high active-materials ratio of 70 wt% and high mass loading of 44.56 mg cm⁻² at 0.53 mA cm⁻². **i** The charge and discharge capacity and coulombic efficiency as a function of cycle number for Li-In|LASI-80Si|TiS₂ASSB at 0.53 cm⁻² for the first ten cycles and 1.06 mA cm⁻² for the subsequent cycles. **j** The comparison of mass loading and areal capacity of TiS₂-based batteries with various SEs, including borohydride SEs, glass, glass-ceramics, LGPS-type and argyrodite-type sulfide SEs, and LASI-80Si in this work. All cells assembled in this work were tested at 30 °C.

ASSB are 175.4, 94.2, and 38 mAh g⁻¹ at 0.76, 3.8, and 7.6 mA cm⁻², respectively. LSPSC ASSB delivers discharge capacities of 175.3, 120.8, and 93 mAh g⁻¹, at 0.76, 3.8, and 7.6 mA cm⁻², respectively. This further corroborates that the superionic conductivity of SE enables the high power of ASSBs. Surprisingly, ASSB using LASI-80Si with an ionic conductivity on par with LSPSC shows much higher discharge capacities of 211.9, 166.8, 138.4 mAh g⁻¹, 0.76, 3.8, and 7.6 mA cm⁻², respectively. It is interesting that the additional capacity contributed by Li₂S and LiI functional phases maintains even at high current densities thanks to the catalytic effect of TiS₂ and LiI on Li₂S/S conversion. Furthermore, the upper-limit current density of LASI-80Si ASSB under various charge–discharge protocols even reaches 24.45 mA cm⁻², as shown in Fig. 5b–d. The asymmetric lithiation/delithiation kinetics between TiS₂ cathode and Li-In alloy anode may result in different upper-limit current densities under distinct charge–discharge protocols. To decouple the factors (e.g., high thickness of composite cathode layer and Li-In alloy) associated with the high mass loading, a low mass loading of 0.51 mg cm⁻² was employed to investigate the intrinsic rate capability and Li-In dendrites suppression ability of the TiS₂|LASI-80Si | Li-In configuration. The current density even reaches 24.45 mA cm⁻², which is 6.4 times larger than the short-circuit current density (3.8 mA cm⁻²) corresponding to the Li-In dendrites penetration through LPSC layer[56]. It is noteworthy that among all reported works (Fig. 5e and Supplementary Table 26), only LGPS sulfide SE enables a highest current density of 17.64 mA cm⁻², which is still one order of magnitude smaller than the rate capability (24.45 mA cm⁻²) enabled by

LASI-80Si in this work. These rate capability performances are well positioned compared to ASSBs testing achieved in the 55–100 °C testing temperature range[13,68].

As a critical indicator, the areal mass loading of cathode active materials directly determines the specific energy of a battery. However, thick electrode also leads to slow Li⁺/e⁻ transportation and sluggish electrochemical kinetics[69]. On one hand, the areal mass loading of TiS₂ was enhanced by increasing the total mass of composite cathode to 50 mg (corresponding to an electrode area of 0.7854 cm²) while maintaining the weight ratio (50 wt%) of TiS₂. As shown in Fig. 5f, a high discharge capacity of 225.9 mAh g⁻¹ is achieved for LASI-80Si ASSB with an areal mass loading of 31.83 mg cm⁻², corresponding to a high areal capacity of 7.19 mAh cm⁻². After several cycles, the specific capacity is stabilized at ~220 mAh g⁻¹ (Fig. 5g) without further decay. On the other hand, the areal mass loading of TiS₂ was increased to 44.56 mg cm⁻², by improving the weight ratio of TiS₂ from 50 to 70 wt%, while keeping the constant total mass (50 mg) of composite electrode. Despite the decreased discharge capacity of 207.6 mAh g⁻¹ (Fig. 5h), the areal capacity reaches an even higher value of 9.26 mAh cm⁻². It is noteworthy that the increased weight ratio of TiS₂ hinders the Li⁺ transport in a certain degree, confirmed by the limited discharge capacity at mild current densities. Although a capacity decay occurs for the initial ten cycles at 0.53 mA cm⁻², the specific capacity is stabilized at ~150 mAh g⁻¹ under an elevated current density of 1.06 mA cm⁻² (Fig. 5i). Figure 5j compares the areal mass loading and capacity of the reported TiS₂-based ASSBs using various SEs (borohydride SEs[58–62], glass[63], glass-

ceramics[70], LGPS-type[66,70], and argyrodite-type[15,67] sulfide SEs) with those of LASI-80Si in this work. The areal mass loading (44.56 mg cm$^{-2}$) and areal capacity (9.26 mAh cm$^{-2}$) in this work exceed the highest level of TiS$_2$ ASSBs with Li$_{6.7}$Si$_{0.7}$Sb$_{0.3}$S$_5$I (23.8 mg cm$^{-2}$ and 5.59 mAh cm$^{-2}$), as summarized in Fig. 5J and Supplementary Table 27. These battery performances are well positioned compared to state-of-the-art literature ASSBs using halide and sulfide double-layer SEs (52.46 mg cm$^{-2}$ and 4.14 mAh cm$^{-2}$ at 50 °C)[71], Ag-C composite anode (31.63 mg cm$^{-2}$ and 4.62 mAh cm$^{-2}$ at 60 °C)[72] and carbon-free silicon anode (60 mg cm$^{-2}$ and 5.7 mAh cm$^{-2}$ at 60 °C)[73].

In summary, a family of argyrodite thioarsenate lithium ionic conductors, Li$_{6+x}$M$_x$As$_{1-x}$S$_5$I (M=Si, Sn), is synthesized and systematically studied to develop a sulfide electrolyte LASI-80Si[32] with multiple functions. By tunning the substitution content of Si$^{4+}$ for As$^{5+}$, an optimized ionic conductivity of 10.4 mS cm$^{-1}$ at 25 °C and a low activation energy of 0.20 eV are achieved for the cold-pressed LASI-80Si pellet. Molecular dynamics analyses with deep-learning potential reveal that additional Li$^+$ ions occupy high-energy sites in LASI-80Si[32] to activate intra-cage concerted migration and a new inter-cage migration pathway with low energy barriers. Furthermore, LASI-80Si shows an improved air stability than phosphorus-based argyrodites (e.g., LPSI and LPSC), in terms of less H$_2$S gas generation and stable crystal structure. This is due to the strong hygroscopicity of LiI and Li$_2$S functional phases and the tight bonding of soft acid As$^{5+}$ to S$^{2-}$ for LASI-80Si. Despite the limited oxidation stability of LASI-80Si[32], when coupled with a Li-In anode and a metal sulfide cathode (i.e., TiS$_2$), the cell exhibits excellent long-term cyclability (62,500 cycles at 2.44 mA cm$^{-2}$). Moreover, the superionic conductivity, interfacial passivation effect and electrochemical Li$^+$-supplement function of LASI-80Si[32] enable TiS$_2$-based ASSBs with high initial CE (99.06% at 0.76 mA cm$^{-2}$), long cycle life (1000 cycles at 0.76 mA cm$^{-2}$ and 62,500 cycles at 2.44 mA cm$^{-2}$), high rate capability (24.45 mA cm$^{-2}$), and high areal mass loading (44.56 mg cm$^{-2}$) and areal capacity (9.26 mAh cm$^{-2}$).

## Methods

### Materials synthesis

All compounds were synthesized by solid-state mechanochemical reactions. Stoichiometric mixtures of Li$_2$S (99.9%, Zhejiang FunLithium New Energy Technology Co., Ltd.), As$_2$S$_3$ (99.9%, Innochem), Si (99.99%, Macklin), LiI (99.95%, Innochem), and elemental sulfur (99.95%, Innochem) were weighed with a total mass of 1 g in an Ar-filled glovebox (O$_2$ < 0.1 ppm, H$_2$O < 0.1 ppm). These mixtures were placed in a zirconia jar filled with zirconia balls (15 large balls with diameter around 9 mm and 30 small balls with diameter around 3 mm) and mechanically milled at 600 rpm for 45 h using a planetary ball milling apparatus. The mass ratio between the mixture and the zirconia ball was 1:40. The ball-milled powders were pelletized at a pressure of 2 tons (~250 MPa), and then sealed in a quartz tube and annealed at 550 °C for 12 h in a muffle furnace which is placed in Ar-filled glovebox (O$_2$ < 0.1 ppm, H$_2$O < 0.1 ppm) to obtain the final products[32]. The preparation of LPSI and LPSC follows the same procedure as mentioned above, but using P$_2$S$_5$ (>99%, Macklin) and LiCl (99.99%, Innochem) as starting materials. The synthesis of LSPSC follows the same procedure as mentioned above, but using P$_2$S$_5$ (>99%, Macklin), Si (99.99%, Macklin), and LiCl (99.99%, Innochem) as starting materials and was annealed at 460 °C for 8 h.

### Physicochemical characterization

X-ray diffraction (XRD) patterns were obtained to characterize the crystal structures of the synthesized materials. The XRD patterns in Fig. 1 were collected in the 2-theta range of 10°–80° using a diffractometer (Beijing Purkinje General Instrument Co., Ltd.) at 36 kV, 20 mA. The Refined diffraction data in Supplementary Fig. 5 was collected in the 2-theta range of 10°–120° by D8 Advance (Bruker Scientific Technology Co., Ltd.). The XRD patterns of LASI-80Si were

Rietveld-refined using the GSAS software package[74,75]. The chemical composition of LASI-80Si was quantified by inductively coupled plasma-atomic emission spectrometry (ICP-AES; Thermo IRIS) for Li, Si, As, high-frequency infrared carbon and sulfur analyzer (YANRUI, CS-320) for S and energy dispersive spectrometer (EDS; HITACHI, SU8100) mapping for I. X-ray photoelectron spectroscopy (XPS; ULVAC-PHI, Inc., PHI 5000 VersaProbe III) with monochromatic Al Kα radiation was used to study the chemical composition of TiS$_2$ composite cathode. First, the whole four-layer pellet/cell was demolding from Swagelok model cell and then the TiS$_2$ composite cathode powders was scraped and harvested from the surface of pellet, in Ar-filled glovebox (O$_2$ < 0.1 ppm, H$_2$O < 0.1 ppm). Then, the powder sample was loaded in sample holder and covered with a sticky tape to avoid air exposure. Finally, the sample holder was transferred from Ar-filled glovebox to equipment for ex situ measurement.

### Ionic conductivity measurements

Ionic conductivity of as-synthesized sulfide SEs was measured by the electrochemical impedance spectroscopy (EIS) measurements. EIS measurements were performed in the frequency range of 1 Hz to 8 MHz and the amplitude of 10 mV using a Zennium pro Electrochemical Workstation. The number of measurement-points above 66 Hz is 10 while that of below 66 Hz is 5. The number of sinewave averages above 66 Hz is 10 while that of below 66 Hz is 5. The testing cell using cold-pressing pellet was fabricated as the following procedures: 100 mg of sulfide electrolyte powders were added into the Swagelok model cell and pressed into a pellet (diameter 1 cm, thickness about 550 μm) with a pressure of 870 MPa (7 tons), directly using two stainless-steel rods as both Li-ion blocking electrodes and current collectors. Finally, the cold-pressed pellet formed in the Swagelok model cell was sealed and placed into a desiccator. To gain the Arrhenius plots, variable-temperature EIS was measured from 0 to 75 °C. The holding time for each temperature is above 4 h. The electronic conductivity of sulfide SEs was measured by direct-current polarization method with a bias voltage of 500 mV at 25 °C.

### Air stability measurements

The stability of sulfide SEs against humid N$_2$ (23–25% RH, 26–29 °C) was tested in a homemade detection system (Supplementary Fig. 9). The humidity was tuned by adjusting the ratio of flow rate of dry N$_2$ gas (99.999%) and humid N$_2$ gas to ensure the sum of these two gas flow rates was 250 mL/min. Here, the humid N$_2$ gas is the N$_2$ gas carrying with water molecules after flowing through the flowmeter B and the humidifier in turn. The humidity of continuous flow of humid N$_2$ gas was measured by a high-precision thermo-hygrometer (179A-THL, Apresys (Shanghai) precision optoelectronics Co., Ltd.). 3 mg sulfide electrolyte powders were weighed and placed into a glass bottle in an Ar-filled glovebox. Then, the sealed bottle was transferred to the outside and connected with the pipeline. The concentration of H$_2$S gas generated from the water reactive SEs was detected and recorded by the H$_2$S gas sensor (JC-AD-2(T), Juchuang Hongye Environmental Technology Co., Ltd.). The total generation amount of H$_2$S was calculated based on the following equation,

$$A\,(\mathrm{cm^3/g}) = \frac{\sum_0^N C_N(\mathrm{ppm})\upsilon(\mathrm{cm^3/min})\Delta t(\mathrm{min})\times 10^{-6}}{M(\mathrm{g})}$$

where $A$ denotes the total/accumulated generation amount of H$_2$S normalized by the weight ($M$) of S atoms in sulfide electrolyte sample, $C_N$ denotes the $N^{th}$ recorded value of H$_2$S concentration, $\upsilon$ is the velocity of nitrogen gas flow and $\Delta t$ is the time interval of recording. The as-synthesized samples were first transferred to the sample holders and covered with a film in Ar-filled glovebox (O$_2$ < 0.1 ppm, H$_2$O < 0.1 ppm). Then, they were simultaneously exposed to laboratory ambience (23–25% RH, 26–29 °C) for 5 min, 10 min and 20 min. Their

XRD patterns before and after exposure were collected at each time block to investigate their structural change and hydrolysis reaction products. These air stability measurements for each sample have been repeated for at least twice.

## Electrochemical stability measurements

The electrochemical stability of four sulfide SEs, namely LPSI, LPSC, LSPSC, and LASI-80Si, was investigated by LSV measurement using an electrochemical workstation (CHI660E, Shanghai Chenhua) with scan rate of $0.1\,mV\,s^{-1}$. The SE and carbon additive (Supper P) were mixed in a 95:5 (wt%) ratio for 30 min using agate mortar and pestle in an Ar-filled glovebox ($O_2 < 0.1$ ppm, $H_2O < 0.1$ ppm) and worked as the cathode material. 100 mg SE powder was cold-pressed into a pellet (~600 μm) under 125 MPa (1 ton) for 1 min, and 10 mg of as-prepared cathode was uniformly spread onto the surface of SE pellet and pressed under a pressure of 874 MPa (7 tons) for 1 min. A piece of Li foil (>99.9%, China Energy Lithium Co., Ltd.) with diameter of 8 mm and thickness of 30 μm was used as the anode and attached on the other side SE layer with a pressure of 125 MPa (1 ton). These LSV measurement results for each sample have been repeated for at least once.

## Preparation of all-solid-state batteries

The composite cathode of Li-In|SE|TiS2 ASSBs consisted of TiS2 powder (99.5%, Macklin), sulfide SEs (one of LPSI, LPSC, LSPSC, and LASI-80Si), which were mixed in a 50:50 (wt%) ratio for 30 min using agate mortar and pestle in Ar-filled glovebox ($O_2 < 0.1$ ppm, $H_2O < 0.1$ ppm). The Li-In alloy anode was prepared by mechanical pressing a piece of In foil (~25 mg, 99.999%, Wanda Scientific Research Metal Materials Co., Ltd) with diameter of 10 mm and thickness of 50 μm and a piece of Li foil (~1 mg, >99.9%, China Energy Lithium Co., Ltd.) with diameter of 8 mm and thickness of 30 μm together. The fabrication of ASSBs has the following procedures: (1) 100 mg sulfide electrolyte powder was added into the Swagelok model cell and pressed under 125 MPa (1 ton) for 1 min to form a flat surface; (2) 5.0 mg composite cathode material was uniformly spread onto the surface of sulfide electrolyte layer and pressed under 873 MPa (7 tons) for 1 min together; (3) In foil was placed on the other side of sulfide electrolyte layer; (4) Li foil was placed on the surface of In foil; (5) finally, the four-layer pellet was formed by pressing at 250 MPa (2 ton) for 1 min. Thus, the four-layer pellet (thickness ~800 μm, mass ~135 mg) cell was sandwiched between two stainless-steel rods as current collector and sealed in the Swagelok model cell with a stacking pressure of ~30 MPa. Galvanostatic charge–discharge was conducted on the LANHE (CT2001A, LAND Electronic Co. Ltd) battery test system. The voltage window was set as 0.9–2.4 V (vs. Li-In/Li+) for Li-In|SE|TiS2cell, and various constant current densities were applied to evaluate the cycling stability and the rate performance of ASSBs at 30 °C. All cell fabrication processes were conducted in an Ar-filled glovebox ($O_2 < 0.1$ ppm, $H_2O < 0.1$ ppm) and all battery test are carried out in the incubator (SHP160, Changzhou Putian Instrument Manufacturing Co., Ltd.) with a constant temperature of 30 °C. It should be noted that the specific capacity reported in this work refers to the mass of active material in composite cathode. The specification of ASSBs fabricated in this work is summarized in Supplementary Table 28. For each item of electrochemical test, at least two Swagelok model cells were tested.

## Molecular dynamics (MD) simulations

The diffusivities and conductivities of the pristine and doped structures are evaluated using MD simulations based on LAMMPS[42]. During all MD simulations, a 4 × 4 × 4 supercell is constructed with 3532 atoms for LASI-80Si or 3328 atoms for LASI system. After a 100 ps warmup in the NPT ensemble to relax lattice parameters, standard MD simulations are executed in the NVT ensemble until the mean square displacement (MSD) of Li ions is greater than 250 Å or the simulation time

is larger than 5 ns, where MSD is defined as:

$$MSD(t) \equiv \left\langle |r(t) - r_0|^2 \right\rangle = \frac{1}{N} \sum_{i=1}^{N} |r^i(t) - r^i(0)|^2$$

where $r^i(t)$ is the position of the ith lithium ion at time $t$, $N$ is the number of lithium ions.

The tracer diffusion coefficient $D_{tr}$ is defined by the mean square displacement (MSD) over time:

$$D_{tr}(T) = \lim_{t \to} \frac{MSD(t)}{6t}$$

where the constant 6 is used for three-dimensional diffusion.

The estimated tracer diffusivity $D$ at temperature $T$, follows the Arrhenius relationship:

$$D(T) = De^{-E_a/k_B T}$$

where $E_a$ is the activation energy and $k_B$ is Boltzmann's constant.

The ionic conductivity $\sigma$ at temperature $T$ was calculated based on the Nernst-Einstein relationship:

$$\sigma(T) = \frac{ne^2 z^2}{k_B T} D(T)$$

where $n$ is the volume density of the diffusing species, $e$ is the unit electron charge, $z$ is the charge of the ionic conductor (1 for lithium ion). The detailed values of tracer diffusion and ionic conductivity at each temperature for LASI, LASI-25Si, LASI-50Si, LASI-80Si are summarized in Supplementary Tables 12–15.

## Potential model training and DFT calculations

In this project, we sampled the training set of force field from AIMD data and obtained machine learning force field with accuracy comparable to DFT. Then, DP-gen[41] software is employed together with DeePMD-kit[40] framework to enhance the efficiency (speed and quantity) of sampling while maintain the accuracy during force field training. Here, it is necessary to clarified that DeePMD-kit is a machine learning force field framework that can integrate the trained force fields into MD software such as LAMMPS. Two potential models are specifically trained in this work. The pristine structure, LASI, is trained individually and all other doped structures, including LASI-25Si, LASI-50Si and LASI-80Si, are trained together based on another model. As similar with the workflow introduced in previous works[41,76], some short Ab initio molecular dynamics (AIMD) trajectories are pre-calculated. For each structure, AIMD simulations at 600, 800, and 1000 K are performed with a duration of 4 ps and a timestep of 2 fs. The initial dataset for DP-Gen framework is extracted by uniform sampling 200 frames from each trajectory. Each DP-Gen iteration consists of three work stages, namely model training, molecular dynamics, and first-principle calculation. During the first stage, several potential models based on initial and generated data from the neural network initialization with distinct random seeds, is trained. Then, MD simulations with trained potential models are performed and structures with large deviations among these models are selected subsequently. Finally, the energy and force properties are calculated by DFT for structures selected from the second stage, and gathered together with previously generated data. Details of DP-Gen settings have been summarized in Supplementary Tables 8 and 9. After several iterations of DP-Gen framework, the dataset for deep potential model training will be expanded and the final training is consequently performed with 2,000,000 steps. The model trained from the final training will be used in MD simulations to obtain diffusivities and conductivities. DFT calculations in this work, including the data generation in DP-Gen

workflow and the NEB method used in energy barrier calculations, are performed using the projector-augmented wave (PAW)[77] approach based on Quantum Espresso 6.8 package[39]. Specifically, a plane wave energy cutoff of 600 eV and a $2 \times 2 \times 2$ k-point mesh is employed. For AIMD simulations, a time interval of 2 fs is adopted. The lattice parameter is fully relaxed at 0 K for all calculations.

## Correlation computation

The self-part and distinct-part van Hove correlation function (i.e., $G_{self}$ and $G_{distinct}$)[78] are defined respectively as follows:

$$G_{self}(r, \Delta t) = \frac{1}{N} \left\langle \sum_{i}^{N} \delta(r - |r_i(t + \Delta t) - r_i(t)|) \right\rangle_t$$

$$G_{distinct}(r, \Delta t) = \frac{1}{4\pi r^2 N \rho} \left\langle \sum_{i}^{N} \sum_{j \neq i}^{N} \delta(r - |r_i(t + \Delta t) - r_j(t)|) \right\rangle_t$$

where $N$ is the number of mobile Li-ions in the unit cell, $r$ is the radial distance, $\delta$ is the one-dimension Dirac delta function, and $\rho$ is the average Li-ion density.

## Reporting summary

Further information on research design is available in the Nature Portfolio Reporting Summary linked to this article.

## Data availability

The data that support the findings of this study are available within the article (and its Supplementary Information files) and from the corresponding author upon reasonable request.

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

## Acknowledgements

This work was supported by Outstanding Youth Fund Project by Department of Science and Technology of Jiangsu Province (Grant No.

BK20220045), Key R&D Project funded by Department of Science and Technology of Jiangsu Province (Grant No. BE2020003), Key Program-Automobile Joint Fund of National Natural Science Foundation of China (Grant No. U1964205), General Program of National Natural Science Foundation of China (Grant No. 51972334), General Program of National Natural Science Foundation of Beijing (Grant No. 2202058), Cultivation Project of Leading Innovative Experts in Changzhou City (CQ20210003), National Overseas High-level Expert Recruitment Program (Grant No. E1JF021E11), Talent Program of Chinese Academy of Sciences, "Scientist Studio Program Funding" from Yangtze River Delta Physics Research Center and Tianmu Lake Institute of Advanced Energy Storage Technologies (Grant No. TIES-SS0001), Joint research program supported by Science and Technology Research Institute of China Three Gorges Corporation (Grant 202103402).

## Author contributions

P.L. and F.W. conceived the research and design the experiments. P.L. performed the experiments. Y.X. performed the theoretical calculations. G.S., D.W., S.W., W.Y., X.Z., J.L., Q.N., S.S., and Z.S. participated in part of the characterization and data analysis. P.L. and F.W. wrote the manuscript, and H.L. and L.C. revised it. All authors discuss the results presented in the manuscript.

## Competing interests

The authors declare no competing interests.
