## [Peer Review File · Nature Communications]

REVIEWER COMMENTS

Reviewer #1 (Remarks to the Author):

In this paper, authors present a comprehensive analysis combining MD simulations and experimental measurements to evaluate the electrochemical stability and performance of multiple argyrodites SEs. A new type of superionic conductor, i.e., LASI-80Si has been proposed with extraordinary ionic conductivity, air stability, long cycle life and superior rate capability. However, as compared to other sulfide SEs (LPSC and LSPSC), LASI-80Si is more vulnerable to oxidation decomposition which may limit their usage with high-voltage cathodes. Following questions should be addressed:

1. All figures looked very blurry, and many texts are too small to read. High resolution figures should be provided.
2. The composition LASI-80Si is a nominal composition (stoichiometric composition for reagents) or the actual composition? More chemical analysis should be done to probe this.
3. The authors shows that the doping limit for Sn and Si compounds are 0.3 and 0.8, respectively. What exactly happened after this doping limit?
4. Will this doping limit change after changing to higher temperatures? Authors should show some data for this.
5. In Fig. 1E, the conductivities of both Sn and Si compounds drop after 0.3 and 0.8, and authors attribute this to doping limits. Is it possible that this is also due to the lack of Li vacancy? Authors should give some discussions and evidence to understand whether it is the lack of Li-vacancy or the doping limits.
6. Authors should give Rietveld refinement results for all compositions they showed in Fig. 1A and B and plotting the lattice constants/volumes as a function of the doping compositions.
7. Authors should give their pellets densities after pressing.
8. AIMD simulations for potential training are only run for 4 ps. This time is too short since most Li-hopping events especially the inter-cage migrations can go beyond this time. Authors should justify that such 4 ps AIMD training dataset is enough. This might be the reason that the computed conductivity (44.5 mS/cm at 300 K) is higher than the experimental value of 10.4 mS/cm.
9. How authors deal with the Si/As and Sn/As disordering in their calculations? There should be many different possible configurations. More details should be given.

10. Authors showed the stability order of LASI > LASI-80Si > LPSC > LPSI in Fig. 3A based on the amount of released H₂S after exposed to moisture. However, such exposure is normalized to per gram of sample. The comparison is not fair because the amounts of sulfur are different in the unit mass of these samples. Therefore, the final amount of the released H₂S could be contributed by the amount of sulfur in these sample. For example, LPSC has 0.597g sulfur per gram of LPSC, whereas LASI-80Si has a 0.431g sulfur per gram of LASI-80Si. LPSC will release more H₂S than LASI-80Si if their stabilities are the same. Authors should normalize their data to per mole to justify their stability order.

Reviewer #2 (Remarks to the Author):

In the submitted manuscript, a novel family of argyrodite thioarsenate lithium ionic conductors, Li_{6+x}M_xAs_{1-x}S₅I₅₋₆₇ (M=Si, Sn) is developed and studied. An ionic conductivity of 10.4 mS/cm is achieved, and arising from this excellent ionic conductivity, all-solid-state (ASS) batteries with the electrolyte, in which TiS₂ is used as the cathode, possess excellent rate capability. By performing systematic analyses as well as literature analysis, the properties and performance of the new electrolytes are rationally explained. However, the use of arsenic with the aim of suppression of H₂S generation is critical for the practical application of the material, and therefore the manuscript is not recommendable for the publication in the Nature Communications without solving following problems.

- 1) The authors explain that the introduction of soft acids can significantly improve the air stability of phosphorus-based sulfide SEs. Compared to As⁵⁺, Sb⁵⁺ should be more soft. Why it was necessary to use As for the electrolyte design instead of Sb? Is there a problem in the preparation of electrolytes including Sb or the ion radius of Sb does not fit those of Sn⁴⁺ and Si⁴⁺?
- 2) The difference in the amount of generated H₂S is small (98.99 cm³/g for LPSC and 91.32 cm³/g for LASI-80Si). It is hardly significant to use highly toxic As to control the generation of H₂S.
- 3) Compared to LPSC and LSPSC, LASI-80Si is less tolerant against oxidation. How does this low electrochemical stability limit the use of high-voltage cathodes? How is the stability of the electrolyte when combined with NMC or LNMO?
- 4) For the publication in the journal with a high impact factor, at least the toxicity/stability or the performance of the electrolytes must be excellent. However, as mentioned in the comment 3), improvement in the former aspect is not notable. In addition, the authors selected TiS₂, having a working voltage of ~1.7V vs. Li-In and a theoretical capacity of 239 mAh/g. Therefore, the energy density of batteries is not remarkably high. This part must be improved for the publication.

5) In the abstract, the electrolytes are characterized as “multi-functional”. In what sense are they multifunctional?

Point-to-Point Response Letter to Reviewers' Comments

Dear Reviewers,

The authors greatly appreciate your insightful comments and careful review on our manuscript. Please find enclosed our point-to-point response to reviewers' comments and revised manuscript, which we would like to submit as the revised version of NCOMMS-22-40650A. This paper has been revised carefully according to the comments of the reviewers. Changes made to the manuscript have been identified by highlighted text in PDF file. The point-to-point responses to reviewers' comments are as following.

Reviewers' comments:

Reviewer #1 (Remarks to the Author):

In this paper, authors present a comprehensive analysis combining MD simulations and experimental measurements to evaluate the electrochemical stability and performance of multiple argyrodites SEs. A new type of superionic conductor, i.e., LASI-80Si has been proposed with extraordinary ionic conductivity, air stability, long cycle life and superior rate capability. However, as compared to other sulfide SEs (LPSC and LSPSC), LASI-80Si is more vulnerable to oxidation decomposition which may limit their usage with high-voltage cathodes. Following questions should be addressed:

We greatly appreciate the positive comments from the reviewer.

1. All figures looked very blurry, and many texts are too small to read. High resolution figures should be provided.

We appreciate the reviewer's valuable suggestion.

We have provided high-resolution (900 dpi) figures in the modified manuscript, and these figures have also been uploaded separately.

2. The composition is a nominal composition (stoichiometric composition for reagents) or the actual composition? More chemical analysis should be done to probe this.

We appreciate the reviewer's suggestion.

The composition of LASI-80Si mentioned in the manuscript is the stoichiometric composition for reagents. Here, we employed multiple methods, such as inductively coupled plasma-atomic emission spectrometry (ICP-AES), carbon and sulfur element analysis and energy dispersive spectrometer (EDS) mapping, to quantify the actual stoichiometric ratio of LASI-80Si.

ICP-AES measurement, as a quantitative and powerful method, was performed to obtain reliable and accurate results. As shown in **Table R1**, the weight percentages of all elements except S and I are very close to the theoretical values deduced from the $\text{Li}_{6.8}\text{Si}_{0.8}\text{As}_{0.2}\text{S}_5\text{I}$ composition. Specifically, the sulfur content is extremely low, which may be induced by the loss of sulfur during preparation and test (e.g., the release of H_2S gas during LASI-80Si dissolution). On one hand,

iodine as the halogen possesses a relatively higher excitation/ionization energy, which may exceed the capability of testing instrument. On the other hand, the loss of iodine (e.g., sublimation at 45°C) during dissolution is also possible. Therefore, these two elements should be further analyzed via other methods.

Table R1. The weight percentage of elements in LASI-80Si obtained from ICP-AES.

Element	Weight%	Theoretical Weight%
Li	12.5	12.7
Si	5.9	6.0
As	4.0	4.0
S	13.5	43.1
I	-	34.1
Totals	100.00	

In terms of the determination of S content, we further performed carbon and sulfur element analyses and obtained reliable result as shown in **Fig. R1**.

Fig. R1 The sulfur content determined by the infrared carbon and sulfur analyzer.

In order to determine the content of I, EDS mapping was performed. To avoid the interference of conductive adhesive, the LASI-80Si sample in pellet form was prepared and characterized, as shown in **Fig. R2**. As summarized in **Table R1**, the weight and atomic percentage of all elements in LASI-80Si except Li are obtained. Although the interference of C has been eliminated, the influence of O is inevitable. As a semi-quantitative method, EDS mapping shows a barely acceptable but imprecise result, compared with the former two methods. Specifically, the measured weight percentages are a little bit different with the theoretical value and the reference value obtained by other two methods.

Fig. R2 EDS mapping results of LASI-80Si pellet.

Table R2. The weight and atomic percentage of elements in LASI-80Si pellet obtained from EDS mapping.

Element	Weight%	Atomic%	Theoretical Weight %
O K	4.24	11.27	
Si K	6.96	10.54	6.0
S K	47.65	63.18	43.1
As L	5.31	3.01	4.0
I L	35.84	12.01	34.1
Totals	100.00		

Therefore, we combine these three measurement results and choose the weight percentages of Li (12.5%), Si (5.9%), As (4.0%) obtained from ICP-AES (**Table R1**), S (35.84%) obtained from carbon and sulfur analyzer (**Fig. R1**), I (35.84%) obtained from EDS mapping (**Table R2**) to calculate the actual stoichiometric ratio of LASI-80Si. The calculated composition $\text{Li}_{6.835}\text{Si}_{0.797}\text{As}_{0.202}\text{S}_{5.125}\text{I}_{1.072}$ is similar to the nominal composition $\text{Li}_{6.8}\text{Si}_{0.8}\text{As}_{0.2}\text{S}_5\text{I}$ based on the reagents.

$$\begin{aligned} &\rightarrow \text{Si: As} = 0.79736: 0.20264 \\ &\rightarrow \text{Li: } 33.7314 \times 0.20264 = 6.83549 \end{aligned}$$

$$\rightarrow \text{S: } 25.2932 * 0.20264 = 5.12553$$

$$\rightarrow \text{I: } 5.28978 * 0.20264 = 1.07195$$

$$\rightarrow \mathbf{Li_{6.835}Si_{0.797}As_{0.202}S_{5.125}I_{1.072}}$$

3. The authors shows that the doping limit for Sn and Si compounds are 0.3 and 0.8, respectively. What exactly happened after this doping limit?

We appreciate the reviewer's comments.

We emphasized the determination of doping limit and the effect induced by excessive doping of Sn and Si in the manuscript, as follows:

“The solid-solution limit can be determined once the impurity phase containing Sn/Si appears. As shown in **Fig. 1A-B**, Sn and Si exist separately in the form of Li_4SnS_4 and Li_4SiS_4 , once reaching the solid-solution limit at $x = 0.3$ and $x = 0.8$, respectively.”

Moreover, the amount of Li_4SnS_4 and Li_4SiS_4 will gradually increase and the face-centered cubic structure will ultimately collapse after exceeding this doping limit.

4. Will this doping limit change after changing to higher temperatures? Authors should show some data for this.

We appreciate the reviewer's valuable suggestion.

As shown in **Fig. R2**, we heat-treated the powder samples after ball-milling at different temperatures higher than 550 °C to synthesize the interested samples. When the temperature is higher than 650 °C, the “LASI-80Si” sample (has the same stoichiometric ratio with LASI-80Si based on the raw materials) is melted and bonded with the Al_2O_3 crucible, which makes the separation of powder sample from crucible very difficult if not impossible.

Fig. R2 The photographs of studied samples after sintering at different temperatures. a. “LASI-80Si” raw materials after ball-milling were sintered at 575, 600, 650 and 750 °C for 12h, respectively. b. “LASI-30Sn” raw materials after ball-milling were sintered at 575 and 600 °C for 12h, respectively.

Consequently, the interested samples sintered at relatively lower temperatures (i.e., 600 and 575 °C) were specially investigated as shown in **Fig. R3** and **Fig. R4**. It should be noted that the doping limit is determined once the impurity phase (e.g., Li_4SnS_4 and Li_4SiS_4) containing Sn/Si rather than any other impurity phases (e.g., Li_2S and LiI) appears. It is obvious that the amount of Li_4SiS_4 and Li_4SnS_4 phases increases as the temperature elevates from 550 to 575 and 600°C. In addition, the amount of LiI and Li_2S impurity phases is also enhanced. Therefore, it can be rationally concluded that the elevated temperature reduces the structural stability of LASI-80Si/LASI-30Sn and induces severe decomposition. Therefore, the doping limit of LASI-xSi and LASI-ySn will be changed or decreased after heating at higher temperature through the degraded structure stability.

Fig. R3 The XRD patterns of interested “LASI-80Si” samples after sintering at different temperatures (550, 575, and 600 °C).

Fig. R4 The XRD patterns of interested “LASI-30Sn” samples after sintering at different temperatures (550, 575, and 600 °C).

- In Fig. 1E, the conductivities of both Sn and Si compounds drop after 0.3 and 0.8, and authors attribute this to doping limits. Is it possible that this is also due to the lack of Li vacancy? Authors should give some discussions and evidence to understand whether it is the lack of Li-vacancy or the doping limits.

We appreciate the reviewer’s valuable comments and suggestions.

It should be noted that the doping limit is merely determined by the impurity phase containing Sn/Si (i.e., doping element), instead of any other impurity phases (i.e., Li_2S and LiI). Moreover, it should be rational that the lack of Li-vacancy may induce the separation of lithium-contained

phases, such as Li_2S and LiI . Consequently, it can be observed that the lithium-contained phases Li_2S and LiI first separate from the cubic structure with the introduction of low-valence doping element Sn/Si. When approaching the solid-solution limit of Sn and Si, the Sn- and Si-contained phases Li_4SnS_4 and Li_4SiS_4 start to separate accompanied with the uninterrupted separation of lithium-contained phases Li_2S and LiI .

In contrast, based on the assumption that the lack of Li vacancy rather than the solid-solution limit results in the drop of ionic conductivities of Sn and Si compounds, this critical point should correspond to the same doping proportion, as the valence of Sn and Si and the amount of interstitial lithium introduced for charge compensation are identical. However, the critical point that the ionic conductivity of Sn and Si compounds starts to decrease is not consistent according to our experimental results (**Fig. 11** or **Fig. R5**). Interestingly, this critical point is coincident with the solid-solution limit (0.3 for Sn and 0.8 for Si compound). Specifically, the expansion or contraction of unit cell induced by Sn or Si substitution in AsS_4 tetrahedrons (**Fig. 1C-D** or **Fig. R6C-D**) will affect the migration of Li^+ as a result of the change of electrostatic interaction and migration distance. Due to the distinct difference on radius for Sn and Si and its effect on ion conduction, this critical/transition point of ionic conductivity is different. On the other hand, the impurity phases (e.g., Li_4SnS_4 or Li_4SiS_4) with low ionic conductivity precipitated at the surfaces or interfaces among fast-ion-conduction particles would decrease the total ionic conductivity measured in experiment.

Therefore, the doping limit rather than the lack of Li-vacancy determines the critical point of ionic conductivity, no matter from the macroscopic phenomena or from the microscopic mechanisms.

Fig. R5 The room-temperature ionic conductivity of $\text{Li}_{6+x}\text{Sn}_x\text{As}_{1-x}\text{S}_5\text{I}$ and $\text{Li}_{6+x}\text{Si}_x\text{As}_{1-x}\text{S}_5\text{I}$ as a function of substitution proportion x .

Fig. R6 (A to D) X-ray diffraction patterns of (A) $\text{Li}_{6+x}\text{Sn}_x\text{As}_{1-x}\text{S}_5\text{I}$ ($x = 0, 0.05, 0.10, 0.15, 0.20, 0.30, 0.40, 0.60$), (B) $\text{Li}_{6+x}\text{Si}_x\text{As}_{1-x}\text{S}_5\text{I}$ ($x = 0, 0.10, 0.20, 0.30, 0.40, 0.50, 0.60, 0.70, 0.80, 0.90, 1.00$), and the enlarged X-ray diffraction patterns at $28^\circ \sim 31^\circ$ for (C) $\text{Li}_{6+x}\text{Sn}_x\text{As}_{1-x}\text{S}_5\text{I}$ ($x = 0, 0.10, 0.30$), (D) $\text{Li}_{6+x}\text{Si}_x\text{As}_{1-x}\text{S}_5\text{I}$ ($x = 0, 0.20, 0.80$).

6. Authors should give Rietveld refinement results for all compositions they showed in Fig. 1A and B and plotting the lattice constants/volumes as a function of the doping compositions.

We appreciate the reviewer's valuable suggestions.

Actually, the XRD patterns shown in **Fig. 1A** and **B** cannot satisfy the requirement of Rietveld refinement (e.g., the intensity of the strongest diffraction peak should be larger than 10,000 counts).

Some representative compositions (LASI, LASI-10Sn, LASI-30Sn, LASI-20Si, LASI-40Si and LASI-80Si) are specially synthesized and characterized to unveil the variation of lattice constants as a function of the doping proportions.

As shown in **Fig. R7**, the high-quality XRD patterns were re-collected for further Rietveld refinements. Sequentially, Rietveld refinements were performed by Fullprof software to obtain the lattice constants. The Rietveld refinement result of undoped sample LASI is shown in **Fig. R8**. The lattice constant of LASI is 10.23377 Å, closing to the theoretical lattice constant 10.23700 Å in numerical value. Following the same procedure, the lattice constants of all representative samples can be obtained, which have been summarized in **Table R3**. Finally, the curves of lattice constants as a function of the doping proportions of Sn and Si have been plotted in **Fig. R9** and **R10**. In general, the lattice constants change linearly with the increase of substitution content in LASI-ySn and LASI-xSi, indicating the consistence with Vegard's law. Coincident with the conclusion mentioned in the manuscript, the introduction of Sn with large ionic radius results in the expansion of crystal lattice. On the contrary, the substitution of Si for As contributes to the contraction of crystal lattice. Specifically, the distinct difference of ionic radius between Sn⁴⁺ (69 pm) and As⁵⁺ (47.5 pm) induces the dramatic variation of lattice constant even at low substitution content. Nevertheless, the similar ionic radius between Si⁴⁺ (40 pm) and As⁵⁺ (47.5 pm) results in the mild change of lattice constants.

Fig. R7 The XRD patterns of LASI-ySn (y = 0, 10 and 30) used for refinement.

Fig. R8 The Rietveld refinement result of undoped sample LASI.

Table R3. The lattice constants of LASI-ySn ($y = 0, 10, 30$) and LASI-xSi ($x = 0, 20, 40, 80$).

Sample	Lattice constant
LASI	10.23377
LASI-10Sn	10.26479
LASI-30Sn	10.29156
LASI-20Si	10.22535
LASI-40Si	10.21465
LASI-80Si	10.21002

Fig. R9 Lattice parameters of LASI-ySn as a function of Sn content.

Fig. R10 Lattice parameters of LASI-xSi as a function of Si content.

7. Authors should give their pellets densities after pressing.

We appreciate the reviewer's valuable suggestion.

We have provided relevant information (i.e., thickness, diameter/area, mass = 0.1 g) of pellets in details, which can be found in **Supplementary Table 3 and Methods part**. Based on **Equation**

(1), we can calculate the pellets densities, which has been added into **Supplementary Table 3** or **Table R5**.

$$\rho = \frac{m}{d \cdot S} \quad (1)$$

Where ρ , m , d and S denote density, mass, thickness and area of pellet, respectively.

Based on **Table R4**, the theoretical density of $\text{Li}_6\text{AsS}_5\text{I}$ solid electrolyte and geometric density of $\text{Li}_6\text{AsS}_5\text{I}$ pellet is 2.50 g/cm^3 and 2.311 g/cm^3 , respectively. The relative density of pellet can reach 92% ($2.311 / 2.50 * 100\% = 92\%$), which is consistent with that of $\text{Li}_6\text{PS}_5\text{Cl}$ pellets reported in the literature (Liu et al., Nano Lett. 2020, 20, 9, 6660–6665).

Table R4. The theoretical density of $\text{Li}_6\text{AsS}_5\text{I}$ solid electrolyte and geometric density of $\text{Li}_6\text{AsS}_5\text{I}$ pellet, and deduced relative density of the pellet.

Electrolytes	Relative molecular mass (g/mol)	Parameter of unit cell (Å)	Molar volume (cm^3)	Theoretical density (g/cm^3)	Thickness (cm)	Geometric density (g/cm^3)	Relative density (%)
$\text{Li}_6\text{AsS}_5\text{I(LASI)}$	403.7971	10.237	161.456	2.50	0.0551	2.311	92.4

According to the data from the last column of **Table R5**, the geometric density roughly increases with the doping amount of Sn, while it decreases with the doping amount of Si. This variation is reasonable as the relative atomic mass of As (74.9216 g/mol) is much larger or smaller than that of Sn (118.71 g/mol) or Si (28.0855 g/mol), given that the volume of unit cell and pellet almost have no change. It should be noted that deviations of geometric densities for some doping compounds may exist, which can be attributed to the bias of thickness and pellet weight. Actually, the thickness at different parts (e.g., central, edge) of pellet may not be consistent, which is related to the cold-pressing technique (uniaxial pressing), equipment, and initial electrolyte powder distribution and etc.. In addition, the pellet mass is regarded to be equal to the mass of electrolyte powders, as the pellet is constricted in the Swagelok model cell and is not convenient to be weighed separately. Nevertheless, the obtained variation trend is reasonable and as expected.

Table R5. The ionic conductivity calculated from the total resistance, thickness and area (0.7854 cm^2) of the cold-pressed pellets of $\text{Li}_{6+x}\text{Sn}_x\text{As}_{1-x}\text{S}_5\text{I}$ ($x=0, 0.05, 0.10, 0.15, 0.20, 0.30, 0.40, 0.60$) denoted as LASI-ySn ($x = y \%$) and $\text{Li}_{6+x}\text{Si}_x\text{As}_{1-x}\text{S}_5\text{I}$ ($x=0, 0.10, 0.20, 0.30, 0.40, 0.50, 0.60, 0.70, 0.80, 0.90, 1.00$) denoted as LASI-ySi ($x = y \%$).

Electrolytes	Substitution proportion x	Thickness (cm)	Total resistance (Ω)	Ionic conductivity (S cm^{-1})	Geometric density (g cm^{-3})
LASI	0	0.0551	17900	3.92×10^{-6}	2.311
	5	0.0617	3940	1.99×10^{-5}	2.064
	10	0.0514	798	8.20×10^{-5}	2.477
	15	0.0527	336	2.00×10^{-4}	2.416
	20	0.0533	344	1.97×10^{-4}	2.389
LASI-xSn	30	0.0546	347	2.00×10^{-4}	2.332
	40	0.0535	816	8.35×10^{-5}	2.380
	60	0.0518	1080	6.11×10^{-5}	2.458
	10	0.0558	8770	8.10×10^{-6}	2.282
LASI-xSi	20	0.0550	2480	2.82×10^{-5}	2.315
	30	0.0594	89.9	8.41×10^{-4}	2.143
	40	0.0604	22.1	3.48×10^{-3}	2.108

50	0.0611	11.4	6.82×10^{-3}	2.084
60	0.0618	12.3	6.40×10^{-3}	2.060
70	0.0612	9.62	8.10×10^{-3}	2.080
80	0.0637	7.8	1.04×10^{-2}	1.999
90	0.0624	42.1	1.89×10^{-3}	2.040
100	0.0613	81700	9.55×10^{-7}	2.077

8. AIMD simulations for potential training are only run for 4 ps. This time is too short since most Li-hopping events especially the inter-cage migrations can go beyond this time. Authors should justify that such 4 ps AIMD training dataset is enough. This might be the reason that the computed conductivity (44.5 mS/cm at 300 K) is higher than the experimental value of 10.4 mS/cm.

We greatly appreciate the reviewer’s comments. To be clearer, we have separated these questions and provided point-to-point responses as follows.

AIMD simulations for potential training are only run for 4 ps. This time is too short since most Li-hopping events especially the inter-cage migrations can go beyond this time.

Actually, the short AIMD trajectories are pre-calculated for subsequent sampling, which is the initial step for potential training. More importantly, the initial dataset is constructed for potential training rather than describing the behavior of atom hopping (e.g., inter-cage migrations).

The relevant description can be found in the “Potential model training and DFT calculations” of computational section in Supplementary Information, which is extracted as follows:

“Similar with the workflow introduced in previous works^{6,7}, some short *Ab initio* molecular dynamics (AIMD) trajectories are pre-calculated. For each structure, AIMD simulations at 600K, 800K and 1000K are performed with a duration of 4ps and a timestep of 2fs. The initial dataset for DP-Gen framework is extracted by uniform sampling of 200 frames from each trajectory. Each DP-Gen iteration consists of three work stages, namely model training, molecular dynamics and first-principle calculation.”

Authors should justify that such 4 ps AIMD training dataset is enough.

Notably, the DP-Gen is started with $3 \times 4 \text{ps} / 2 \text{fs} = 6000$ structures at three different temperatures (600K, 800K and 1000K), which is much larger than that of the literature (590 structures, Huang, J. *et al. Deep potential generation scheme and simulation protocol for the Li₁₀GeP₂S₁₂-type superionic conductors. J Chem Phys 154, 094703, (2021)*).

In addition, after several iterations, the training dataset is significantly expanded, which has been described as follows:

“After several iterations of DP-Gen framework, the dataset for deep potential model training will be expanded and the final training is consequently performed with 2,000,000 steps (i.e., 2 ns).” Further details about DP-Gen iterations have been summarized in **Supplementary Table 87** and **9**.

The accuracy of the machine learning force field model and the stability of MD simulation is greatly improved after several DP-Gen iterations. Specifically, compared with the energy and force generated from DFT method, the deviation of those generated from DP-Gen scheme has been summarized in **Supplementary Table 7 or Table R6**. This ignorable deviation demonstrates the high accuracy of the molecular dynamics (MD) with deep-learning potential method, in addition to its superior efficiency to DFT.

Table R6. Root-mean square errors of the energy (meV/atom) and force (meV/Å) of LASI and LASI-80Si on the whole dataset generated from DP-Gen scheme.

Root-mean square errors	LASI	LASI-80Si
Energy (meV/atom)	1.06	1.36
Force (meV/Å)	38.8	45.6

In addition, it is highly possible that the MD simulation with maximum duration of 2 ns (2,000,000 steps) at 500K~1000K (**Supplementary Table 8 and 9**) can capture inter-cage Li-hopping events.

This might be the reason that the computed conductivity (44.5 mS/cm at 300 K) is higher than the experimental value of 10.4 mS/cm.

Based on some ideal assumptions, such as ignoring the effect of grain boundary on ionic conductivity and assumption of single phase, most of the theoretical calculation results obtained by AIMD or MD with deep-learning potential method are rarely consistent with experimental results in numerical value. Consequently, it is acceptable that the experimentally-measured ionic conductivity (10.4 mS/cm) is different from the theoretical value (44.5 mS/cm). Moreover, MD with deep-learning potential method employed in this work not only shows comparable accuracy with that of DFT method (**Table R6**), but also significantly improves the computational temperature range, cell size and simulation time compared to AIMD method. In addition, the activation energy predicted by MD method is not only consistent with NEB method but also similar to the experimental result, which supports the reliability of machine learning force field model.

9. How authors deal with the Si/As and Sn/As disordering in their calculations? There should be many different possible configurations. More details should be given.

We greatly appreciate the reviewer's comments.

The initial cell structures of LASI-25Si, LASI-50Si, and LASI-75Si used in AIMD calculation are generated based on the LASI cell after 1/4, 2/4 and 3/4 Si substitution at the As (4b) site, respectively. In order to achieve charge conservation, the additional Li cations were randomly placed into Li1(48h) sites in four cages centered on S(4c) sites, without distinguishing different 48h sites in the same cage. Consequently, there are 16 ($C_4^1 \cdot C_4^1$), 36 ($C_4^2 \cdot C_4^2$) and 16 ($C_4^3 \cdot C_4^3$) different LASI-25Si, LASI-50Si and LASI-75Si unit-cell structures, respectively, after ignoring the symmetry.

As the probability of Si locates in As/Si (4b) site is 4/5 for LASI-80Si, the probability of finding one of LASI, LASI-25Si, LASI-50Si, LASI-75Si and LASI-100Si in LASI-80Si system (containing four As/Si (4b) sites) is $1/625 \left(\left(\frac{1}{5}\right)^4\right)$, $16/625 \left(C_4^1 * \left(\frac{1}{5}\right)^3 * \left(\frac{4}{5}\right)^1\right)$, $96/625 \left(C_4^2 * \left(\frac{1}{5}\right)^2 * \left(\frac{4}{5}\right)^2\right)$, $256/625 \left(C_4^3 * \left(\frac{1}{5}\right)^1 * \left(\frac{4}{5}\right)^3\right)$, and $256/625 \left(\left(\frac{4}{5}\right)^4\right)$. Consequently, the 4x4x4 LASI-80Si supercell containing 64 unit cells can be regarded as random distribution and connection of 0

(~1/625*64) LASI, 2 (~16/625*64) LASI-25Si, 10 (~96/625*64) LASI-50Si, 26 (~256/625*64) LASI-75Si and 26 (~256/625*64) LASI-100Si unit cells. Notably, this is a convenient method to construct the structure of LASI-80Si, as the substitution proportion (i.e., 80% for LASI-80Si) adopted in experiment does not conform to the usual practice in theoretical calculation, which causes a dilemma (exceeding our calculation ability) in calculation if strictly constructing a perfect structure of LASI-80Si.

10. Authors showed the stability order of LASI > LASI-80Si > LPSC > LPSI in Fig. 3A based on the amount of released H₂S after exposed to moisture. However, such exposure is normalized to per gram of sample. The comparison is not fair because the amounts of sulfur are different in the unit mass of these samples. Therefore, the final amount of the released H₂S could be contributed by the amount of sulfur in these sample. For example, LPSC has 0.597g sulfur per gram of LPSC, whereas LASI-80Si has a 0.431g sulfur per gram of LASI-80Si. LPSC will release more H₂S than LASI-80Si if their stabilities are the same. Authors should normalize their data to per mole to justify their stability order.

We appreciate the reviewer's valuable comments.

As sulfide samples serve as the only sulfur source, the sulfur content in specific sample directly determines the amount of H₂S gas. Consequently, it is not reasonable to make a comparison based on the amount of H₂S normalized by the sample weight since different samples may contain distinct amount of sulfur. Indeed, we completely agree with the reviewer's viewpoint and have employed a more reasonable normalization when calculating the amount of H₂S in our previous work (Lu, P. et al., *Advanced Materials* **33**, 2100921, (2021)). To the best of our knowledge, this normalization is currently the most reasonable method and also have been adopted by other groups (Hayashi, A. et al., *Nat Commun* **10**, 5266, (2019)). Therefore, this normalization procedure is also exploited in this work.

The relevant descriptions have been added in Supplementary Information (or "Methods" section in the revised manuscript) as following:

"The total generation amount of H₂S was calculated based on the following equation,

$$A(\text{cm}^3/\text{g}) = \frac{\sum_0^N C_N(\text{ppm})v(\text{cm}^3/\text{min})\Delta t(\text{min}) \times 10^{-6}}{M(\text{g})}$$

where A denotes the total/accumulated generation amount of H₂S normalized by the weight (M) of S atoms in sulfide electrolyte sample, C_N denotes the N^{th} recorded value of H₂S concentration, v is the velocity of nitrogen gas flow and Δt is the time interval of recording."

Reviewer #2 (Remarks to the Author):

In the submitted manuscript, a novel family of argyrodite thioarsenate lithium ionic conductors, Li_{6+x}M_xAS_{1-x}S₅I (M=Si, Sn) is developed and studied. An ionic conductivity of 10.4 mS/cm is achieved, and arising from this excellent ionic conductivity, all-solid-state (ASS) batteries with the electrolyte, in which TiS₂ is used as the cathode, possess excellent rate capability. By performing systematic analyses as well as literature analysis, the properties and performance of

the new electrolytes are rationally explained. However, the use of arsenic with the aim of suppression of H₂S generation is critical for the practical application of the material, and therefore the manuscript is not recommendable for the publication in the Nature Communications without solving following problems.

We greatly appreciate the positive comments and valuable suggestions from the reviewer.

1. The authors explain that the introduction of soft acids can significantly improve the air stability of phosphorus-based sulfide SEs. Compared to As⁵⁺, Sb⁵⁺ should be more soft. Why it was necessary to use As for the electrolyte design instead of Sb? Is there a problem in the preparation of electrolytes including Sb or the ion radius of Sb does not fit those of Sn⁴⁺ and Si⁴⁺?

We sincerely appreciate your comments.

Based on the empirical hard and soft acid and base (HSAB) theory, Sb⁵⁺ should be softer than As⁵⁺ as a result of its large ion radius. However, as shown in **Fig. R11**, the Li-M-S compound for Sb⁵⁺ as the central cation delivers much lower hydrolysis reaction than that of As⁵⁺, indicating that compound contained Sb⁵⁺ are more instable to air/moisture than that of As⁵⁺ (Zhu et al., *Angew. Chem. Int. Ed.* 2020, 59, 17472 –17476). Therefore, Sb⁵⁺ is not a better alternative for As⁵⁺ in terms of its air stability. In addition, the atomic mass of Sb (121.76 g/mol) is much higher than As (74.92 g/mol), which indicates that the volume of Sb-based sulfide electrolyte powders with same weight will be lower than that of As counterpart. For instance, when their mass is identical, Sb-based sulfide electrolyte may have less contact area with active materials and reduced membrane thickness compared to As counterpart.

The ionic radius of Sb⁵⁺ (60 pm) is much larger than that of As⁵⁺ (46 pm). Consequently, Sn⁴⁺ (69 pm) would be more adequate to replace Sb, while Si⁴⁺ (40 pm) is more propriate to replace As⁵⁺ (46 pm), according to the matching degree of ionic radius.

Figure 2. Hydrolysis reaction energy of sulfides, including 46 binary M-S (grey circles), 52 ternary Li-M-S (green triangles) and 65 Na-M-S (orange triangles), as a function of cation M. More-negative reaction energy indicates worse moisture stability (moisture sensitive), whereas more-positive reaction energy indicates better moisture stability (moisture stable). The horizontal dashed lines correspond to the hydrolysis reaction energy of Li_2S (green) and Na_2S (orange), respectively.

Fig. R11. The calculated moisture stability of Li-M-S ternary sulfide compound as a function of central cation M.

- The difference in the amount of generated H_2S is small ($98.99 \text{ cm}^3/\text{g}$ for LPSC and $91.32 \text{ cm}^3/\text{g}$ for LASI-80Si). It is hardly significant to use highly toxic As to control the generation of H_2S .

We greatly appreciate the reviewer's comments.

Based on the results from **Supplementary Table 17** or **Table R7** and comparing the total amount of generated H_2S of LPSI, LPSC and LASI, it can be deduced that As^{5+} has a positive effect on improving the air stability. However, Si^{4+} as a hard acid is not a good dopant to enhance air stability according to the calculation results in **Fig. R12**, in spite of its slightly better stability than P^{5+} . Therefore, although the optimized composition LASI-80Si owns the highest ionic conductivity and decreased toxicity, its air stability is compromised by the large substitution amount of Si.

Table R7. The total generation amount of H_2S , the peak value of the generation rate and its corresponding peak position of LPSI, LPSC, LSPSC, LASI and LASI-80Si sulfide SEs.

Electrolyte	Total amount of H_2S ($\text{cm}^3 \text{ g}^{-1}$)	Peak value of the generation rate ($\text{cm}^3 \text{ g}^{-1} \text{ min}^{-1}$)	Peak position (min)
LPSI	105.35	25.56	0.3333
LPSC	98.99	49.95	0.4167
LASI	75.07	37.19	0.4167
LASI-80Si	91.32	39.89	0.4167

Figure 2. Hydrolysis reaction energy of sulfides, including 46 binary M–S (grey circles), 52 ternary Li–M–S (green triangles) and 65 Na–M–S (orange triangles), as a function of cation M. More-negative reaction energy indicates worse moisture stability (moisture sensitive), whereas more-positive reaction energy indicates better moisture stability (moisture stable). The horizontal dashed lines correspond to the hydrolysis reaction energy of Li_2S (green) and Na_2S (orange), respectively.

Fig. R12 The calculated moisture stability of Li-M-S ternary sulfide compounds as a function of central cation M.

Furthermore, it should be noted that not all sulfur anions coordinate with central cations (i.e., P, As, Si, Sn) in argyrodite structure, compared to Thio-LISICON structure (e.g., Li_3PS_4 , Li_4SnS_4 and etc.). Some sulfur anions coordinate with Li^+ and form cage-like Li_6S octahedrons, which may be a critical instability factor that results in a much higher H_2S generation amount for all compositions. In addition to the sulfur coordination environment, the numeric value of measured H_2S amount is associated with the testing conditions. Compared to other testing systems and normalization method reported in literatures (e.g., Adv. Energy Mater. 2020, 2002861), which may be affected by the large volume of enclosed desiccator (e.g., 2.5 L) and sample forms/weight (pellet, 200 mg, **Fig. R13**), the numeric value of H_2S gas measured in a tiny bottle (e.g., 25 mL) and normalized by the sulfur mass (powder, 5 mg) for this work is relatively higher.

Fig. R13 H_2S amount generated when $\text{Li}_{6.6}\text{Ge}_{0.6}\text{Sb}_{0.4}\text{S}_5\text{I}$ and $\text{Li}_6\text{PS}_5\text{Cl}$ were exposed to air.

Actually, arsenic solid electrolytes, such as Li_3AsS_4 , $\text{Li}_{3.334}\text{Ge}_{0.334}\text{As}_{0.666}\text{S}_4$, $\text{Li}_{3.833}\text{Sn}_{0.833}\text{As}_{0.166}\text{S}_4$, $\text{Li}_{3.875}\text{Sn}_{0.875}\text{As}_{0.125}\text{S}_4$, Na_3AsS_4 and $\text{Na}_3\text{P}_{0.62}\text{As}_{0.38}\text{S}_4$ have been actively studied in recent years (Journal of Materials Chemistry A, 2, 10396 (2014); Energy Environ. Sci., 2014, 7, 1053-1058; Adv. Mater. 2021, 2100921; Adv. Mater. 2017, 29, 1605561). Arsenic ion as a soft acid has displayed significant effect on improving the ionic conductivity and air stability of sulfide solid electrolytes. Apart from experimental studies, theoretical calculation on arsenic compounds (e.g., Li/Na-As-S, Li/Na-M-As-S-X) also attracted numerous attentions (Journal of The Electrochemical Society, 163 (9) A2079-A2088 (2016); Angew. Chem. Int. Ed. 2020, 59, 17472 –17476). Moreover, some arsenic compounds (e.g., As_2S_3 and As_4S_4) naturally exist and have some special application from ancient times to nowadays. Although arsenic compounds may not be suitable for large-scale application, scientific researches on them are still significant and should not be interrupted. Similarly, although lead (Pb) is one of the most toxic and accumulative heavy-metal elements known, it is still widely used in chemical, cable, batteries and radioactive protection industries.

3. Compared to LPSC and LSPSC, LASI-80Si is less tolerant against oxidation. How does this low electrochemical stability limit the use of high-voltage cathodes? How is the stability of the electrolyte when combined with NMC or LNMO?

We greatly appreciate the reviewer's comments.

Actually, we have specially investigated the performance of Ni90@LZO/LASI-80Si/Li-In all-solid-state battery before choosing TiS_2 cathode as an alternative. Here, Ni90 represents $\text{LiNi}_{0.9}\text{Mn}_{0.05}\text{Co}_{0.05}\text{O}_2$ and LZO is the Li_2ZrO_3 coating layer. As shown in **Fig. R14** and **R15**, the initial coulombic efficiency of this battery is as low as 50.86% and that of the subsequent cycles is merely around 95%, much lower than 99.9%. Although the initial charge capacity is as high as 254.8 mAh g^{-1} , the capacity degrades rapidly with the increase of cycle number. Based on the long-term cycle stability of TiS_2 /LASI-80Si/Li-In and the superior reduction stability of LASI-80Si, it can be deduced that the LASI-80Si/Li-In interface at anode side is very stable. Consequently, it can be speculated that the unstable Ni90@LZO/LASI-80Si interface at cathode side contributes to the low coulombic efficiency and degraded discharge capacity. This phenomenon has also been observed in a similar battery configuration (i.e., $\text{LiCoO}_2/\text{Li}_{16.7}\text{Si}_{10.7}\text{Sb}_{0.3}\text{S}_5/\text{Li-In}$ all-solid-state battery), as shown in **Fig. R16** (Zhou et al., J. Am. Chem. Soc. 2019, 141, 19002–19013). However, when matching with low-voltage cathodes (e.g., TiS_2), this unstable phenomenon disappeared. In addition, experimental results have shown that the LZO coating layer can ensure the Ni90@LZO/LPSC interfacial stability and long-term cyclability. Therefore, it can be concluded that this unstable issue originates from the low oxidation stability of the iodine-based argyrodite materials (e.g., LASI-80Si). Its electrochemical oxidation decomposition can not be passivated or interrupted by the interface layer consisting of the decomposition products, thus resulting in the extremely low coulombic efficiency (i.e., charge capacity > discharge capacity). Moreover, the increased interfacial resistance induced by the poor ionic conductivity and uninterrupted growth of interfacial layer leads to the extremely low capacity and its degradation with cycle number.

Fig. R14 The charge-discharge profiles of Ni90@LZO/LASI-80Si/Li-In all-solid-state battery.

Fig. R15 The long-term cyclability of Ni90@LZO/LASI-80Si/Li-In all-solid-state battery.

Fig. R16 The long-term cyclability of LiCoO₂/Li_{6.7}Si_{0.7}Sb_{0.3}S₅I/Li-In all-solid-state battery.

4. For the publication in the journal with a high impact factor, at least the toxicity/stability or the performance of the electrolytes must be excellent. However, as mentioned in the comment 3), improvement in the former aspect is not notable. In addition, the authors selected TiS₂, having a working voltage of ~1.7V vs. Li-In and a theoretical capacity of 239 mAh/g. Therefore, the energy density of batteries is not remarkably high. This part must be improved for the publication.

We appreciate the reviewer's comments.

Although the working voltage of TiS₂ is relatively low, its theoretical energy density (active material's level) is comparable to the commercialized LiCoO₂, as verified by the following equations (average voltage * nominal capacity = gravimetric energy density).

$$\begin{aligned}\text{LiCoO}_2: 3.92 \text{ V} * 135 \text{ mAh g}^{-1} &= 529.2 \text{ Wh kg}^{-1} \\ \text{TiS}_2 (1.5\text{V}-3\text{V}): 2.22 \text{ V} * 235.8 \text{ mAh g}^{-1} &= 523.5 \text{ Wh kg}^{-1}\end{aligned}$$

LiNi_{0.9}Mn_{0.05}Co_{0.05}O₂ (Ni90) as a promising Ni-rich cathode shows a much higher energy density of 807.7 Wh kg⁻¹ than that of LiCoO₂.

$$\text{LiNi}_{0.9}\text{Mn}_{0.05}\text{Co}_{0.05}\text{O}_2 (\text{Ni}90): 3.81 \text{ V} * 212 \text{ mAh g}^{-1} = 807.7 \text{ Wh kg}^{-1}$$

Here, we choose the sulfur-rich cathode TiS₄ as a representative case to further demonstrate the promise of superionic conductor LASI-80Si in high-energy-density application. As shown in **Fig. R17** and **Fig. R18**, TiS₄ cathode exhibits much higher energy density than that of Ni90, in spite of its low working voltage.

$$\begin{aligned}\text{TiS}_4 (0.6\text{V}-3\text{V}): 1.94 \text{ V} * 940 \text{ mAh g}^{-1} &= 1823.6 \text{ Wh kg}^{-1} \\ \text{TiS}_4 (1.5\text{V}-3\text{V}): 2.03 \text{ V} * 600 \text{ mAh g}^{-1} &= 1218 \text{ Wh kg}^{-1}\end{aligned}$$

In addition to matching with the low-voltage cathodes, recent experimental results have shown that employing a configuration with double-layer solid electrolyte (i.e., LiNi_{0.85}Co_{0.1}Mn_{0.05}O₂/Li₂In_xSc_{0.666-x}Cl₄/Li_{6.7}Si_{0.7}Sb_{0.3}S₅I/Li-In) is a promising strategy to achieve high output voltage. Consequently, it can be deduced that this strategy should also be applicable to LASI-80Si if the requirement of high-voltage cathodes was necessary.

Therefore, the combination with low-voltage TiS₂ cathode in this work merely validate the promise of LASI-80Si in extreme-fast-charging and high-loading applications, which does not impede further investigation or application of superionic conductor LASI-80Si as a promising solid electrolyte in high-energy-density and even high-voltage scenarios. Indeed, we have validated the successful combination of LASI-80Si solid electrolyte with the low-cost, high-energy-density FeS₂ pyrite cathode (1.85 V * 800 mAh g⁻¹ = 1480 Wh kg⁻¹) in our recent work (Lu et al., Adv. Funct. Mater. 2022, 2211211).

In terms of the energy density at the battery level, it is better to be evaluated in soft pouch cell with thin separator layer (~30 um) rather than the Swagelok model cell with thick separator layer (~500 um). However, numerous challenges still exist (e.g., large interface resistance and

stacking pressure, inconsistency of performance and short-circuit issue) to be dealt with from the lab model cell to the large-capacity soft pack cell. In addition, Li-In is often accepted as a standard/reference anode in sulfide-based all-solid-state battery, due to the short-circuit issues aroused by lithium anode and the infancy of other promising anodes. Nevertheless, the compromised output voltage induced by Li-In alloy anode is still a challenge to be overcome in the future. Consequently, this work only focuses on the investigation of the physicochemical properties of LASI-80Si solid electrolyte and preliminary verification of its enormous potential in battery performance.

Fig. R17 The long-term cyclability of $\text{TiS}_4/\text{LASI-80Si}/\text{Li-In}$ all-solid-state battery within a voltage range of 1.5-3V (vs. Li^+/Li).

Fig. R18 The long-term cyclability of $\text{TiS}_4/\text{LASI-80Si}/\text{Li-In}$ all-solid-state battery within a voltage range of 0.6-3V (vs. Li^+/Li).

5. In the abstract, the electrolytes are characterized as “multi-functional”. In what sense are they multifunctional?

We do appreciate the reviewer's comments.

In addition to the functions as a superionic conductor and electron-blocking separator, LASI-80Si also has a unique lithium-ion-supplementary effect, which has been specially proved and discussed at the second paragraph of the “**Electrochemical performance**” section. As shown in **Fig. R19 B-D**, the initial charge capacity and initial coulombic efficiency (black lines) of LASI-80Si all-solid-state battery are much higher than those of LPSC and LSPSC. The slight decomposition of LASI-80Si only occurs at the first cycle, which can be identified from the charge-discharge profiles in **Fig. R19D**. These two phenomena are coincident with the features of lithium-ion supplement. Therefore, LASI-80Si have demonstrated multifunctional features.

More relevant descriptions can be found as follows:

“Besides, the relatively stronger intensity of Li_2S for $\text{TiS}_2/\text{LASI-80Si}$ composite electrode than that of $\text{TiS}_2/\text{LSPSC}$ counterpart after cycling indicates a slight electrochemical decomposition of LASI-80Si into Li_2S and LiI products. Given the high theoretical capacity of Li_2S (1166 mAh g^{-1}) and LiI (200 mAh g^{-1}), and the catalytic effect^{50,51} of TiS_2 and LiI on $\text{Li}_2\text{S}/\text{S}$ conversion reaction, Li_2S and LiI functional phases are speculated to bring about the improved specific charge capacity, initial CE and rate capability. To prove this assumption, 3 wt% Li_2S or LiI , a similar content in as-synthesized LASI-80Si, was first introduced during the preparation of TiS_2/LPSC composite electrode. Both Li_2S - and LiI - incorporated composite cathodes exhibit an enhanced specific charge capacity (Fig. 4E), initial CE (**Supplementary Table 22**) and rate capability (**Supplementary Figure 15**).”

“Therefore, it is reasonable to conclude that the in-situ formed and uniformly distributed Li_2S and LiI functional phases in LASI-80Si serve as lithium supplements by de-lithiation to replenish Li^+ sacrificed in interphase formation or trapped in TiS_2 interlayer during charging. This contributes to the large specific charge capacity (228.7 mAh g^{-1} at 0.1C and 216.9 mAh g^{-1} at 1C), high initial CE (100.33% at 0.1C and 97.57% at 1C) and excellent rate capability of LASI-80Si ASSB.”

Fig. R19 (Fig. 4 in the manuscript) Charge-discharge profiles. The charge-discharge profiles of TiS_2 | SE | Li-In ASSBs with (A) LPSI (0.02C), (B) LPSC, (C) LSPSC, and (D) LASI-80Si sulfide SEs, respectively. Due to the extremely low ionic conductivity of LPSI ($\sim 10^{-6} \text{ S cm}^{-1}$) compared with the other three sulfide SEs, TiS_2 | LPSI | Li-In ASSB can only cycle at low current rates ($\leq 0.02\text{C}$) and delivers a low specific capacity. (E) Comparison of 1st-cycle charge-discharge profiles for TiS_2 composite cathodes without and with $\text{Li}_2\text{S}/\text{LiI}$ additives, including $0.5\text{TiS}_2+0.5\text{LPSC}$ (black), $0.5\text{TiS}_2+0.47\text{LPSC}+0.03\text{Li}_2\text{S}$ (blue), $0.5\text{TiS}_2+0.47\text{LPSC}+0.03\text{LiI}$ (red) and $0.5\text{TiS}_2+0.47\text{LPSC}+0.06\text{LiI}$ (cyan). (F) Comparison of 1st-cycle charge-discharge profiles for various TiS_2 composite cathodes, including $0.5\text{TiS}_2+0.5\text{LPSC}$ (black), $0.5\text{TiS}_2+0.44\text{LPSC}+0.03\text{Li}_2\text{S}+0.03\text{LiI}$ (blue), and $0.5\text{TiS}_2+0.5\text{LASI-80Si}$ (red). The numerical value before each component represents its weight ratio in composite cathodes.

REVIEWERS' COMMENTS

Reviewer #1 (Remarks to the Author):

Authors have shown their efforts and fully addressed my comments therefore I recommend for publication in Nature Communications.

Reviewer #2 (Remarks to the Author):

The authors made a lot of efforts for improving the quality of paper, which is highly appreciated. The answer to the 1st comment (relation between the softness of ion and the air stability of resulting material) would be useful information for the readers. It is recommended to include these information in the introduction. Instead of summarizing results in the 3rd paragraph of introduction, it should be nice to summarize the strategy of materials design.

Dear Editor and reviewers,

The authors greatly appreciate your insightful comments and careful review on our manuscript. Please find enclosed our point-to-point response to reviewers' comments and revised manuscript, which we would like to submit as the revised version of NCOMMS-22-40650. This paper has been revised carefully according to the comments of the reviewers. Changes made to the manuscript have been identified by highlighted text in MS Word. The point-to-point responses to reviewers' comments are as following.

Reviewer #1 (Remarks to the Author):

Authors have shown their efforts and fully addressed my comments therefore I recommend for publication in Nature Communications.

We greatly appreciate the recommendation for publication by reviewer 1.

Reviewer #2 (Remarks to the Author):

The authors made a lot of efforts for improving the quality of paper, which is highly appreciated.

We greatly appreciate the approval of reviewer 2.

1. The answer to the 1st comment (relation between the softness of ion and the air stability of resulting material) would be useful information for the readers. It is recommended to include these information in the introduction.

We appreciate the valuable suggestion from reviewer 2.

The relevant information has been added into the second paragraph of the introduction. The description is as follows:

“It is noteworthy that a softer acid may not definitely lead to a better moisture stability. The moisture stability of sulfide SEs with these soft acids as the only central cations follows the order of $\text{In}^{3+} > \text{As}^{5+} > \text{Sn}^{4+} > \text{Ge}^{4+} > \text{Sb}^{5+}$, according to the thermodynamics analysis²⁹.”

2. Instead of summarizing results in the 3rd paragraph of introduction, it should be nice to summarize the strategy of materials design.

We appreciate the valuable suggestion from reviewer 2.

In order to emphasize the strategy of materials design according to your suggestion, we have modified the second paragraph in introduction as follows:

“Recent experimental results reveal that the partial substitution of soft acids including Ge^{4+} ¹⁸, Sn^{4+} ¹⁹, As^{5+} ²⁰, Sb^{5+} ²¹, In^{3+} ²² for hard acid P^{5+} can enhance the air stability of phosphorus-based sulfide SEs, based on the hard and soft acid base (HSAB) theory. However, these phosphorus-based sulfide SEs still suffer from irreversible structural degradation and H_2S gas release. Phosphorus-free Thio-LISCON (lithium ion superionic conductor) sulfide SEs, such as Li_4SnS_4 ²³, $0.4\text{LiI}-0.6\text{Li}_4\text{SnS}_4$ ²⁴, $\text{Li}_{4-x}\text{Sn}_{1-x}\text{As}_x\text{S}_4$ ^{20,25}, $\text{Li}_{4-x}\text{Sn}_{1-x}\text{Sb}_x\text{S}_4$ ^{26,27}, and Li_3SbS_4 ²⁸, have been designed through complete substitution to exhibit moisture stability and even recoverability after heat treatment. It is noteworthy that a softer acid may not definitely lead to a better moisture stability. The moisture stability of sulfide SEs with these soft acids as the only central cations follows the order of $\text{In}^{3+} > \text{As}^{5+} > \text{Sn}^{4+} > \text{Ge}^{4+} > \text{Sb}^{5+}$, according to the thermodynamics analysis²⁹. However, the ionic conductivities at 25 °C of this kind of SEs are generally lower than 1 mS cm^{-1} . Recently, Zhou et al.¹⁵ developed phosphorus-free argyrodites, $\text{Li}_{6+x}\text{M}_x\text{Sb}_{1-x}\text{S}_5\text{I}$ ($\text{M} = \text{Si}, \text{Ge}, \text{Sn}$), which displayed high conductivity over 10 mS cm^{-1} at 25 °C for cold-pressed pellet and improved air stability³⁰ compared with $\text{Li}_6\text{PS}_5\text{I}$.”